# Multilayered control of splicing regulatory networks by DAP3 leads to widespread alternative splicing changes in cancer

Jian Han [1✉], Omer An [1], Xi Ren[1], Yangyang Song [1], Sze Jing Tang [1], Haoqing Shen [1], Xinyu Ke[1], Vanessa Hui En Ng [1], Daryl Jin Tai Tay [1], Hui Qing Tan [2], Dennis Kappei [1,3,4], Henry Yang [1] & Leilei Chen [1,4,5✉]

The dynamic regulation of alternative splicing requires coordinated participation of multiple RNA binding proteins (RBPs). Aberrant splicing caused by dysregulation of splicing regulatory RBPs is implicated in numerous cancers. Here, we reveal a frequently overexpressed cancer-associated protein, DAP3, as a splicing regulatory RBP in cancer. Mechanistically, DAP3 coordinates splicing regulatory networks, not only via mediating the formation of ribonucleoprotein complexes to induce substrate-specific splicing changes, but also via modulating splicing of numerous splicing factors to cause indirect effect on splicing. A pan-cancer analysis of alternative splicing across 33 TCGA cancer types identified DAP3-modulated mis-splicing events in multiple cancers, and some of which predict poor prognosis. Functional investigation of non-productive splicing of *WSB1* provides evidence for establishing a causal relationship between DAP3-modulated mis-splicing and tumorigenesis. Together, our work provides critical mechanistic insights into the splicing regulatory roles of DAP3 in cancer development.

[1] Cancer Science Institute of Singapore, National University of Singapore, Singapore 117599, Singapore. [2] Department of Physiology, Yong Loo Lin School of Medicine, National University of Singapore, Singapore 117549, Singapore. [3] Department of Biochemistry, Yong Loo Lin School of Medicine, National University of Singapore, Singapore 117596, Singapore. [4] NUS Centre for Cancer Research, Yong Loo Lin School of Medicine, National University of Singapore, Singapore 117594, Singapore. [5] Department of Anatomy, Yong Loo Lin School of Medicine, National University of Singapore, Singapore 117594, Singapore. ✉email: csihj@nus.edu.sg; polly_chen@nus.edu.sg

RNA-binding proteins (RBPs) participate in every aspect of RNA processing and regulation, from RNA transcription, splicing, cleavage, and polyadenylation to RNA modifications, degradation, transportation, and translation. Each of these processes involves multiple RBPs which form regulatory networks to execute the dynamic control of the transcriptome and proteome complexity. The spatiotemporal regulation of RNA processing by RBPs is vital for normal development and physiology, so that any disruption in RNA processing may lead to human diseases[1]. Aberrant RNA splicing is frequently observed in almost all types of cancers and each cancer hallmark may be affected by aberrant splicing[2]. Alternative splicing in cancer cells can switch a gene from a tumor-suppressive or non-oncogenic isoform to an oncogenic isoform. For instance, mutually exclusive alternative splicing of the *PKM* pre-mRNA generates either an exon 9 inclusion isoform *PKM1*, or an exon 10 inclusion isoform *PKM2*. Through upregulation of heterogeneous nuclear ribonucleoproteins (hnRNPs) that repress *PKM* exon 9 inclusion, cancer cells express high level of the *PKM2* isoform to maintain aerobic glycolysis[3]. Alternative splicing is also a prevalent mechanism to regulate gene expression by altering the stability and degradation of mRNA transcripts. Inclusion of "poison exons" that contain premature termination codons can trigger nonsense-mediated decay (NMD), a translation-dependent RNA surveillance process that degrades mRNAs[4]. For example, mutation of *SRSF2* alters its RNA-binding recognition and thereby promotes mis-splicing and NMD of *EZH2*, results in impaired hematopoietic differentiation in myelodysplastic syndromes (MDS) development[5].

Next-generation sequencing reveals that aberrant splicing often occurs at a genome-wide scale in cancer cells, with numerous genes alternatively spliced to drive cancer initiation and progression. Although *cis*-acting mutations can lead to deficient pre-mRNA splicing by inactivating a splice site within the pre-mRNA, most cancer-driven or associated splicing events are resulted from altered expression and/or function of splicing factors and regulators due to *trans*-acting mutations or other genomic and epigenomic mechanisms. For example, genomic amplification of a proto-oncogene and splicing factor *SRSF1* is observed in many cancer types, which leads to a malignant transformation of rodent fibroblasts to sarcomas by inducing an oncogenic isoform of *RPS6KB1*[6]. In addition to identifying changes in splicing regulatory *cis*-elements in RNA, it is therefore critical to delineate the precise mechanisms underpinning misregulation of *trans*-acting splicing regulators that bind to RNA and their functional relevance to cancer.

RBPs can form complex and interlaced RNA processing regulatory networks via direct binding to RNA or protein–protein interactions with other RBPs. Different RBPs may share a high degree of sequence recognition similarity and therefore work in a cooperative or competitive mode[7,8]. Many RBPs, and particularly splicing factors, auto- or cross-regulate expression of themselves or other RBPs through alternative splicing coupled with NMD (AS-NMD) respectively, increasing the complexity of splicing regulatory networks[9,10]. In this study, we identify death-associated protein 3 (DAP3) as an RNA splicing regulator and uncovered its multilayered control of cancer transcriptome. DAP3 binds extensively to endogenous RNAs in vivo and demonstrates an RNA-binding preference for coding sequences (CDS) with a significant enrichment in splicing-associated motifs such as the 5' splice site consensus sequence and SR-protein hexamer motif. DAP3 can not only facilitate the association of splicing factor proline and glutamine rich (SFPQ) and Non-POU domain containing octamer binding (NONO) with target RNAs for splicing modulation, but also modulate the expression of numerous splicing factors via AS-NMD, leading to global changes in splicing. Notably, such widespread splicing changes modulated by DAP3 can be observed in multiple cancer types and are of clinical relevance and prognostic values. By further investigations of functional importance of DAP3-driven splicing events in cancer using in vitro and in vivo models, our study provides critical mechanistic insights into the role of DAP3 in cancer development as a critical regulator of RNA splicing.

## Results

**RNA-binding landscape of DAP3 indicates its potential role in RNA splicing.** It has previously been shown that DAP3 interacts with RNA editing enzymes adenosine deaminases acting on RNA (ADARs) and functions as a potent repressor of adenosine-to-inosine (A-to-I) RNA editing in cancer cells[11]. Surprisingly, however, DAP3 does not directly bind to the regions proximal to editing sites[11] and the RNA-binding landscape of DAP3 remains largely unclear so far. Herein, we performed a comprehensive transcriptome-wide analysis of RNA-binding sites of DAP3 using our eCLIP-Seq data[11] (GEO accession number: GSE144318) and found that DAP3 binds extensively to endogenous RNAs in vivo with 9699 genes showed DAP3-bound peaks in both biological duplicates (Supplementary data 1). No presence of four selected RBPs including SFPQ, NONO, U2AF35, and U2AF65 in the DAP3 eCLIP elutes confirmed that our eCLIP experiment could specifically pull down DAP3-bound RNAs (Supplementary Information; Supplementary Fig. 1a, b). Further analysis showed that ~70% of DAP3-bound peaks were mapped to exons (37,624 out of 53,596 peaks and 45,575 out of 64,555 peaks in biological duplicates DAP3-1 and DAP3-2, respectively) (Fig. 1a; Supplementary data 1). Of note, DAP3 demonstrated a strong RNA-binding preference for CDS appearing slightly skewed towards 5' end of CDS (Fig. 1b). Moreover, although DAP3 binds predominately to protein-coding RNAs (~80% of eCLIP peaks), ~6% of eCLIP peaks was mapped to non-coding RNAs (ncRNAs), such as two well-known ncRNAs *NEAT1* and *MALAT1* (Fig. 1c). We also observed that DAP3 can bind to multiple regions of one gene, such as 5'UTR, first intron and CDS of *ARHGEF16* gene and different sites within *TARDBP* 3'UTR (Fig. 1c). We also conducted a comparative analysis of gene structure between DAP3-bound and unbound genes and did not observe any significant difference in the number of exons or isoforms between two groups of genes (Supplementary Fig. 2a, b). Further, the de novo motif analysis of DAP3-binding sites by HOMER[12] showed the top three enriched DAP3 RNA-binding motifs: GAAGAAGAU, C(A/U)(A/U)C repeats, and AGGUAAGU (Fig. 1d). The most enriched motif GAAGAAGAU contains a representative GAAGAA hexamer, which is an exonic splicing enhancer (ESE) essential for constitutive and alternative splicing[13]. The presence of AGGUAA GU motif, the 5' splice site consensus sequence in vertebrates[14], indicates DAP3 could bind to the exon–intron splice junctions (Supplementary Fig. 3a, b). We next selected two target transcripts *SCYL1* and *ATAD3A* for the RNA electrophoretic mobility shift assay (REMSA), due to the presence of top ranked DAP3 RNA-binding motifs in their eCLIP peaks (Supplementary Fig. 3c). We confirmed the direct binding of DAP3 to these target RNAs in vitro (Fig. 1e).

Gene ontology (GO) enrichment analysis revealed that DAP3 tends to bind and affect genes associated with RNA processing (e.g., splicing and degradation), DNA metabolic process and mitotic cell cycle regulation (Fig. 1f). All these observations indicate a potential role of DAP3 in modulating RNA splicing.

**DAP3 modulates widespread alternative splicing.** To investigate the role of DAP3 in RNA splicing modulation, we quantified splicing changes caused by *DAP3* depletion in two esophageal squamous cell carcinoma (ESCC) cell lines EC109 and KYSE180, from previously published total RNA-Seq data[11] using rMATS

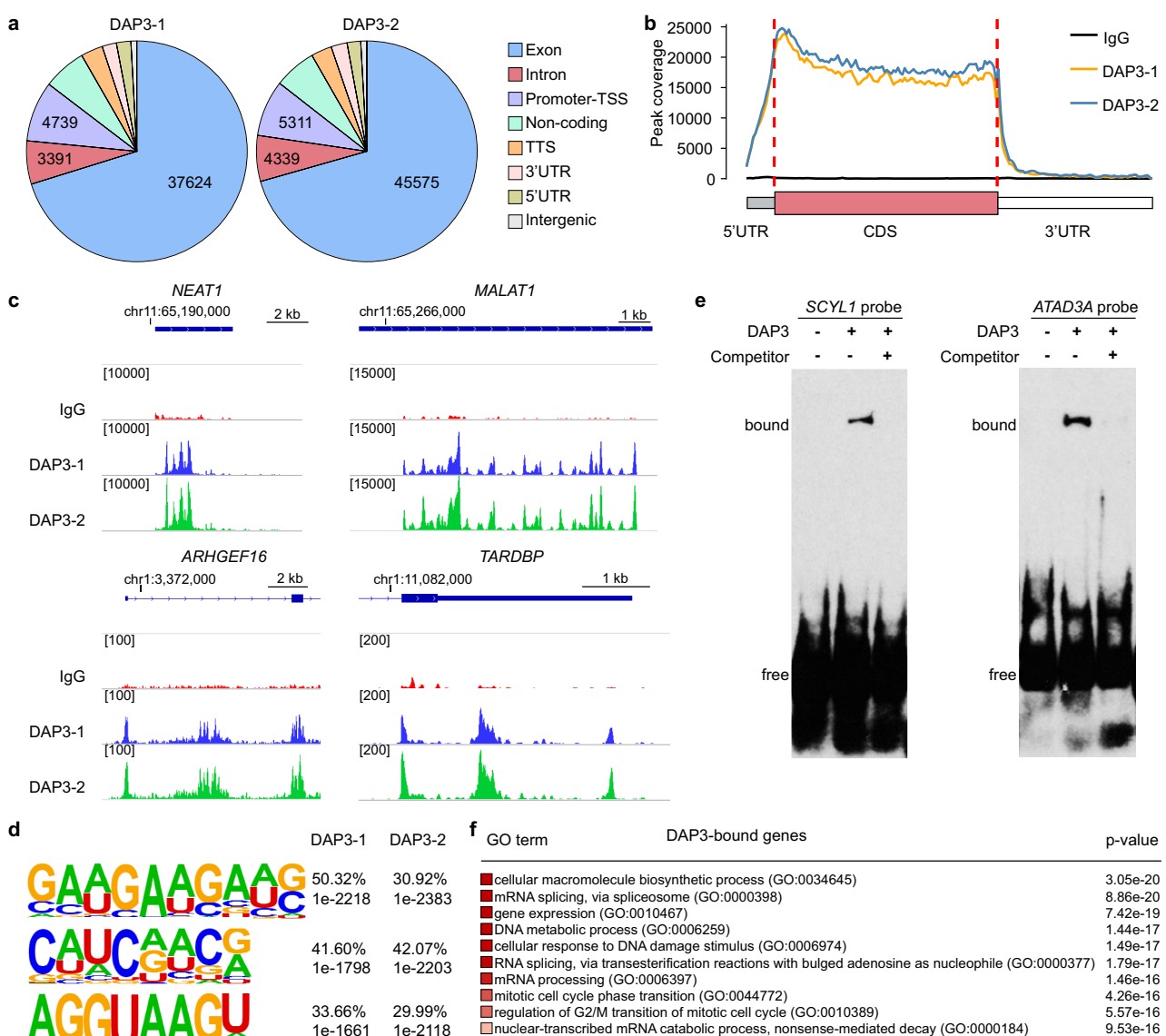

**Fig. 1 RNA-binding landscape of DAP3 by eCLIP-Seq. a** The distribution of the DAP3 eCLIP peaks in the human genome from biological duplicates (DAP3-1 and DAP3-2). TSS transcription start site, TTS transcription termination site, UTR untranslated region. **b** The binned DAP3 eCLIP peak coverage across all expressed genes in EC109 cells. The 5′UTR, CDS, and 3′UTR of each gene are split into 13, 100, and 70 bins, respectively. CDS coding sequence. **c** Integrated genome viewer (IGV) browser tracks of the DAP3 eCLIP peaks spanning the genomic loci of *NEAT1* and *MALAT1*, 5′ region of *ARHGEF16*, and 3′ region of *TARDBP* gene. Detailed information about all significantly enriched eCLIP peaks can be found in Supplementary data 1. **d** Top three most significantly enriched de novo sequence motifs in the DAP3-binding peaks using HOMER[12]. The percentage of peaks containing the discovered motifs and the *p*-values of the motifs calculated by binomial test against the random genomic background were shown. **e** REMSA analysis of the binding of DAP3 to two RNA probes that were generated from DAP3-binding sequences within *SCYL1* and *ATAD3A* transcripts. A 100-fold molar excess of nonbiotinylated RNA probes was used as competitor. **f** Gene ontology analysis of DAP3-bound genes. The top 10 most significantly enriched biological processes are shown. The significance of enrichment for GO sets were evaluated by the WebGestalt[39] portal default hypergeometric test. Source data are provided in Source Data file.

pipeline[15]. Compared to the scramble controls, alternative splicing events in *DAP3*-depleted cells [|Δpercent spliced-in (ΔPSI)| ≥ 10%, false discovery rate (FDR) < 0.05, and splice junction read coverage ≥20] were defined as DAP3-modulated splicing events. Intriguingly, we identified 7400 and 11,820 DAP3-modulated splicing events in 3262 and 4582 genes in EC109 and KYSE180 cells, respectively (Fig. 2a; Supplementary data 2 and 3). Approximately half of the DAP3-modulated splicing events are skipped exon (SE), followed by mutually exclusive exon (MXE; ~20%), alternative 5′ splice site (A5SS; ~10%), alternative 3′ splice site (A3SS; ~10%), and intron retention (IR; ~10%) in both cell lines (Fig. 2a; Supplementary data 2 and 3).

DAP3 was inclined to repress exon skipping (58% and 71% of DAP3-repressed vs. 42% and 29% DAP3-promoted SE events in EC109 and KYSE180, respectively), intron retention (72% and 77% vs. 28% and 23%), and usage of distal A5SS (65% and 58% vs. 35% and 42%) and A3SS (65% and 65% vs. 35% and 35%) events (Fig. 2b). Based on the observation that half of the DAP3-modulated splicing events belongs to SE, we further analyzed whether there is a preferential regulation of constitutively included, excluded or alternatively spliced exons by DAP3. SE events identified in the EC109 and KYSE180 scramble control cells (108,399 and 110,895 in EC109 and KYSE180, respectively) were stratified into groups based on their PSI values (Supplementary

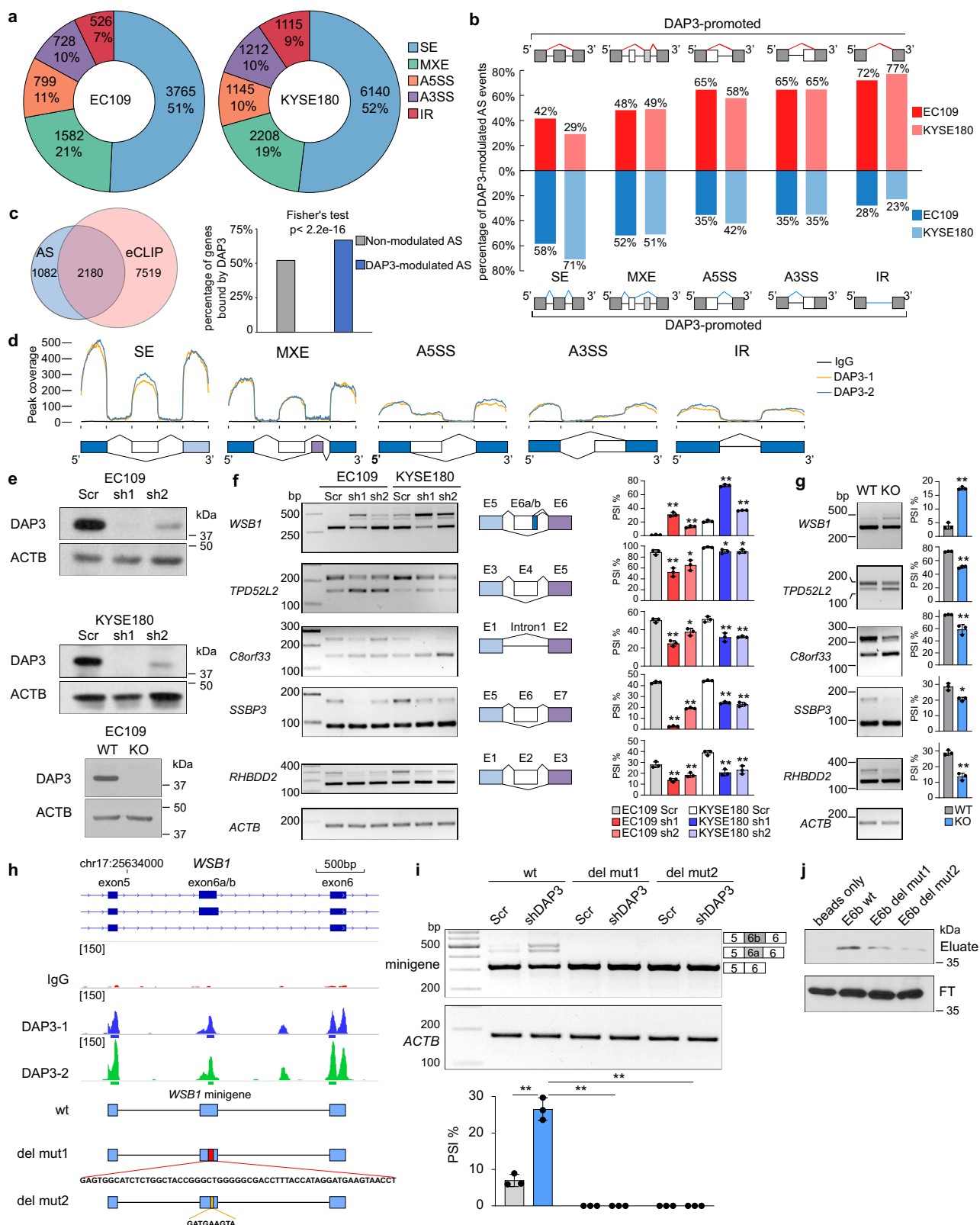

Fig. 4a). We found that less than 1% of SE events in "PSI = 0%" (i.e., constitutively excluded) group and less than 3% of SE events in "PSI = 100%" (i.e., constitutively included) group were modulated by DAP3; while 15–25% of SE events detected in the remaining groups with basal PSI values ranging from 20 to 80% were modulated by DAP3. These findings indicate that compared to constitutively included or excluded exons, those with basal PSI

values ranging from 20 to 80% are preferentially affected by DAP3.

A combined analysis of the DAP3 eCLIP-Seq and RNA-Seq datasets of EC109 cells showed that 67% (2180/3262) or 52% (5121/9828) of genes with or without DAP3-modulated alternative splicing were bound by DAP3, respectively, demonstrating a significant enrichment of DAP3-binding peaks in genes undergoing

**Fig. 2 *DAP3* depletion leads to widespread splicing changes in cancer cells. a** Pie charts showing the distribution of each type of significantly altered splicing events in *DAP3*-depleted EC109 and KYSE180 cells compared to the scramble controls detected by rMATS[15]. SE skipped exon, MXE mutually exclusive exons, A5SS alternative 5′ splice site, A3SS alternative 3′ splice site, IR intron retention. **b** Percentage of DAP3-modulated splicing events demonstrating increased or decreased PSI upon *DAP3* depletion in EC109 and KYSE180 cells. **c** Venn diagram showing the numbers of genes which underwent DAP3-modulated splicing and genes containing DAP3 eCLIP peaks in EC109 cells. Bar chart showing the percentage of genes with or without DAP3-modulated splicing that are bound by DAP3 (two-sided Fisher's Test). **d** The binned DAP3 eCLIP peak coverage across the splice junctions of five types of DAP3-modulated splicing events in EC109 cells. **e** Western blot analysis of DAP3 protein expression in the indicated *DAP3* knockdown (KD, sh1, and sh2), knockout (KO) and their control [scramble (Scr) and wildtype (WT)] samples. **f, g** Semiquantitative RT-PCR analyses of five randomly selected DAP3-modulated splicing events. **h** IGV browser tracks of the DAP3 eCLIP peaks spanning exon 5–6 and intervening introns of *WSB1* gene. Significant peaks are marked by blue and green bars. Schematic diagram illustrates the genomic fragments inserted into the wildtype (wt) and mutant *WSB1* splicing minigenes. del mut1, lacking a 58 bp DAP3-binding sequence in exon E6a/b; del mut2, lacking a 9 bp DAP3-binding motif in exon E6a/b. **i** Semiquantitative RT-PCR analyses of splicing changes of exogenous *WSB1* transcripts derived from the indicated minigenes in Scr control and *DAP3*-KD cells. **j** RNA pulldown assay detecting the binding of DAP3 to *WSB1* exon E6b wt, del mut1, and del mut2 RNA probes. WB analysis of DAP3 proteins in RNA pulldown (eluate) products and flow-through (FT) fractions. **f, g, i** Data are represented as mean ± s.d. of *n* = 3 biologically independent samples. Statistical significance is determined by unpaired, two-tailed Student's *t*-test (**p* < 0.05, ***p* < 0.01). Exact *p*-values and source data are provided in Source Data file.

DAP3-modulated splicing (Fisher's test, *p* < 2.2e-16; Fig. 2c). Our further randomization analysis confirmed that such a binding enrichment was not due to the difference in expression level or sample size between these two groups of genes (Supplementary Information; Supplementary Fig. 4c and d). Based on these observations, binding of DAP3 to its target RNA transcripts may be required for at least a subset of DAP3-modulated alternative splicing events. To further understand whether the binding affinity of DAP3 to RNA targets is associated with the strength of its splicing regulation, we divided DAP3-bound genes based on the fold enrichment of their eCLIP peaks into four groups (≥4, ≥8, ≥12, and ≥16 fold) and examined the percentage of bound genes with or without DAP3-modulated splicing among all DAP3-bound genes. We found that the proportion of bound genes undergoing DAP3-modulated splicing remains within the range of 20–25% among different groups (Supplementary Fig. 4e), indicating an increased binding affinity of DAP3 does not potentiate its splicing regulation.

Further analysis of the eCLIP peak coverage for all five types of DAP3-modulated splicing events indicated a strong tendency of binding to exons rather than introns (Fig. 2d). The HOMER[12] motif enrichment analysis of sequences from DAP3-modulated cassette exons identified a top ranked AGGUAAGU motif (Supplementary Fig. 4f), which matches with the third enriched motif from DAP3-bound sequences (Fig. 1d) and the second enriched motif identified from sequences of DAP3-binding peaks located within the region from the upstream to downstream constitutive exon of DAP3-modulated splicing event (Supplementary Fig. 4g).

Next, a total of 20 DAP3-modulated splicing events were experimentally validated in *DAP3*-knockdown (KD) cells (Figs. 2e, f, 4b, and 5g). Five of which were chosen for further verification in *DAP3*-knockout (KO) cells (Fig. 2e and g). To provide experimental evidence that DAP3 binding to target RNA is required for its splicing regulation, we selected one of the DAP3 target genes *WSB1* for further investigation. We first constructed a wildtype (wt) minigene consisting of *WSB1* exons 5–6 and intervening introns. Based on the identified DAP3 eCLIP peaks on *WSB1* gene, we generated two deletion mutant minigenes by deleting a 58 bp DAP3-binding sequence on *WSB1* exon E6a/b (del mut1) or a 9 bp GATGAAGTA DAP3-binding motif (del mut2) (Fig. 2h). We found depletion of DAP3 led to the inclusion of previously unannotated exon E6a and E6b in the wt minigene (Fig. 2i), consistent with the splicing change of endogenous *WSB1* upon DAP3 depletion (Fig. 2f). However, such splicing changes were not observed in *WSB1* transcripts derived from del mut1 or del mut2 minigene upon DAP3 knockdown (Fig. 2i). Next, RNA pulldown assays using RNA probe consisting of the *WSB1* wt exon E6b (E6b wt), the 58nt DAP3-binding sequence-depleted

exon E6b (E6b del mut1), or the 9nt GATGAAGTA binding motif-depleted exon E6b (E6b del mut2) further confirmed the deleted sequences were required for DAP3 binding (Fig. 2j). These findings suggest that for genes undergoing DAP3-modulated splicing such as *WSB1*, the binding of DAP3 to its target RNA is required for its splicing regulation.

Moreover, given that one-third (1082/3262) of genes with DAP3-modulated alternative splicing events was not bound by DAP3, it is possible that DAP3 may also modulate splicing through RNA-binding-independent mechanisms. Because A-to-I RNA editing could potentially affect splicing[16], we further intersected DAP3-regulated editing sites with DAP3-modulated splicing events and found only 1–2% of splicing events having editing sites detected in the region from the upstream to downstream constitutive exon of an alternatively spliced event (Supplementary Fig. 4h and i). Although this does not exclude the possibility that DAP3-regulated RNA editing in nascent RNAs could impact pre-mRNA splicing, editing-mediated alternative splicing does not stand as a dominant mechanism of DAP3-mediated splicing regulation.

GO analysis suggested that genes undergoing DAP3-modulated alterative splicing were functionally enriched in pathways associated with DNA replication, mitotic cell cycle phase transition, and RNA processing (Supplementary Fig. 5). Together, these findings provided solid evidence that DAP3 is a critical regulator of widespread alternative RNA splicing and dramatically reshapes transcriptome in cancer cells by altering thousands of alternative splicing events.

**DAP3 facilitates the association of splicing regulators SFPQ and NONO with target RNAs for splicing modulation**. To dissect the mechanism by which DAP3 functions as an RBP to modulate splicing, we first explored whether DAP3 interacts with other RBPs, such as splicing factors, to facilitate or block their binding to target RNAs. We performed immunoprecipitation coupled to mass spectrometry (IP-MS) in combination with SILAC (stable isotope labelling with amino acids in cell culture) to detect and quantify DAP3-interacting proteins (Fig. 3a). We identified two well-known RNA splicing regulators SFPQ and NONO among the top DAP3-interacting proteins in both forward and reverse SILAC experiments (Fig. 3b; Supplementary data 4). Reciprocal co-IP analysis further confirmed that DAP3 interacted with SFPQ and NONO (Fig. 3c–e), and their interactions were not abolished by RNase A treatment prior to the co-IP experiments (Fig. 3f). Of note, by examining three exemplary target transcripts *WSB1*, *SSBP3*, and *RHBDD2*, we noticed that opposite to the effect of *DAP3* depletion, overexpression of SFPQ

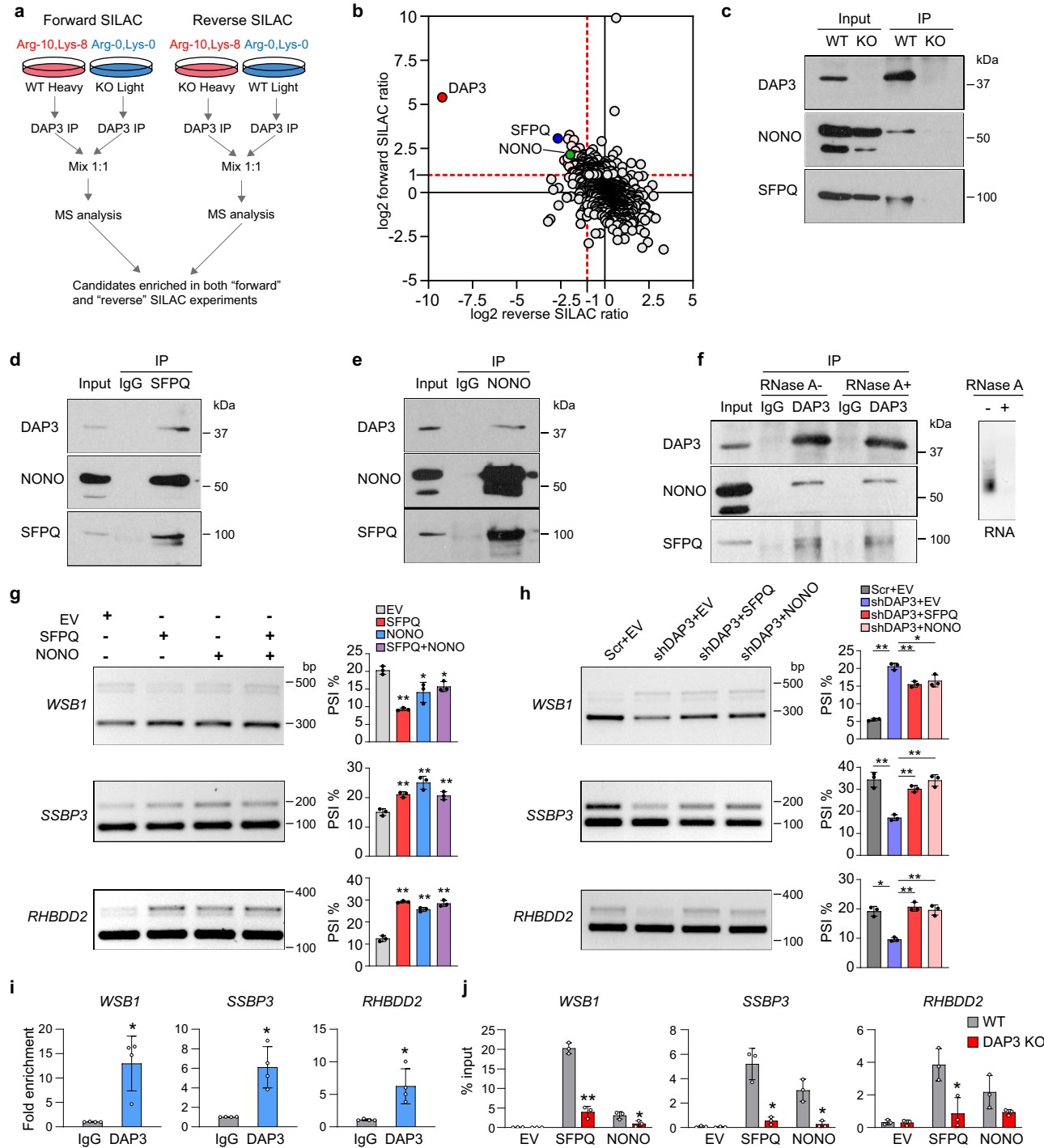

and NONO alone or together significantly promoted skipping of *WSB1* exon E6a/E6b, whereas repressed skipping of *SSBP3* exon 6 and *RHBDD2* exon 2 (Fig. 3g; Supplementary Fig. 6a). Moreover, introduction of SFPQ and NONO into *DAP3*-depleted cells reversed splicing changes of *WSB1, SSBP3*, and *RHBDD2* caused by *DAP3*-KD (Fig. 3h; Supplementary Fig. 6b). These observations prompted us to examine whether DAP3 might form a splicing regulatory complex with SFPQ and NONO to interact with mRNA transcripts and modulate RNA splicing cooperatively. To this end, after experimental verification of the association of DAP3 with *WSB1, SSBP3*, and *RHBDD2* (Fig. 3i; Supplementary Fig. 7), we found that in the absence of DAP3, the association of SFPQ and NONO to these transcripts were

significantly reduced (Fig. 3j; Supplementary Fig. 6c). Altogether, we demonstrated that DAP3 complexes with the splicing regulators SFPQ and NONO in an RNA-independent manner and facilitates their association with target RNA transcripts for splicing modulation.

**DAP3 modulates splicing of numerous splicing factors.** As described above, approximately one-third of transcripts which undergo DAP3-modulated alternative splicing do not have DAP3-binding sites detected by eCLIP-Seq (Fig. 2c). We next explored whether DAP3 could modulate alternative splicing via other mechanisms independent of RNA binding. We first looked

**Fig. 3 DAP3 facilitates the binding of splicing factors SFPQ and NONO to target RNAs. a** Flowchart of the immunoprecipitation coupled to mass spectrometry (IP-MS) in combination with SILAC (stable isotope labelling with amino acids in cell culture) to detect DAP3 interactors. **b** Scatterplot of DAP3-immunoprecipitated proteins retrieved from both forward and reverse SILAC-based IP-MS. DAP3 and several top interactors (e.g., SFPQ and NONO) are highlighted. **c** Co-IP analysis of protein extracts from the WT and *DAP3*-KO EC109 cells. IP was performed with a DAP3 antibody, followed by western blot analysis of DAP3-pulldown products using DAP3, SFPQ, and NONO antibodies. **d, e** Co-IP analysis of protein extracts from EC109 cells. IP was performed with a SFPQ (**d**) or NONO (**e**) antibody, followed by western blot analysis of SFPQ or NONO-pulldown products using the indicated antibodies. IgG antibody was used as negative control. **f** Co-IP analysis of protein extracts from EC109 cells. RNase A treated (RNase A+) or untreated (RNase A−) protein lysates were used for IP using DAP3 antibody, followed by western blot analysis of DAP3-pulldown products using the indicated antibodies. IgG antibody was used as negative control. Agarose gel demonstrating the successful digestion of total RNA. **g** Semiquantitative RT-PCR analyses of the indicated splicing events upon overexpression of SFPQ or NONO alone or together in EC109 cells. **h** Semiquantitative RT-PCR analyses of the indicated splicing events upon overexpression of SFPQ or NONO in *DAP3*-KD EC109 cells. **i** RIP-qPCR analysis of the association between DAP3 protein and the indicated mRNA transcripts (*WSB1*, *SSBP3*, and *RHBDD2*). Data are represented as mean ± s.d. of $n = 4$ biologically independent samples. **j** RIP-qPCR analysis of the binding of SFPQ or NONO to the indicated mRNA transcripts in *DAP3*-KO and WT EC109 cells that were transfected with Flag-tagged SFPQ or NONO, respectively. RIP was conducted using anti-Flag M2 beads. **g, h, j** Data are represented as mean ± s.d. of $n = 3$ biologically independent samples. **g–j** Statistical significance is determined by unpaired, two-tailed Student's *t*-test (*$p < 0.05$, **$p < 0.01$). Exact *p*-values and source data are provided in Source Data file.

into over 300 genes which are involved in mRNA processing and have their splicing modulated by DAP3 (Supplementary Fig. 5), including genes encoding the spliceosomal small nuclear ribonucleoproteins (*SNRPA, SNRPA1, SNRPB, SNRPE, SNRPG*, and *SNRNP27*), RNA-binding motif proteins (*RBM3, RBM4, RBM5, RBM6, RBM7, RBM10, RBM15, RBM23, RBM28*, and *RBM39*), Serine/Arginine-rich (SR) splicing factors (*SRSF1, SRSF3, SRSF5, SRSF7, SRSF10*, and *SRSF11*), heterogeneous nuclear ribonucleoproteins (*HNRNPC, HNRNPD, HNRNPK, HNRNPH1*, and *HNRNPH2*), and DEAD-box helicases (*DDX5, DDX19B, DDX23, DDX39A, DDX42*, and *DDX46*), and found that many of which form functional protein association networks associated with various steps of RNA processing such as splicing and poly-adenylation (Fig. 4a). We then went on to show that depletion of *DAP3* truly promoted skipping of *RBM6* exon 6, *NSRP1* exon 2, *AKAP17A* exon 5b, *FMR1* exon 12, and *HNRNPH1* exon 4, as well as the usage of a distal 3′SS of *SNRPB* exon 7 and a distal 5′SS of *PRPF4B* exon 12 and *TIA1* exon 8; while repressed skipping of *RBM4* exon 3 and the usage of a distal 3′SS of *TIAL1* exon 3 (Fig. 4b). It is well-known that splicing factors can autoregulate their own expression or cross-regulate other splicing factors through alternative splicing coupled nonsense-mediated decay (AS-NMD)[9,10]. Hypothetically, among 10 selected DAP3-modulated splicing factors, DAP3 can trigger the NMD of *RBM6, HNRNPH1, AKAP17A*, and *TIA1* by introducing a premature termination codon (PTC). As expected, cycloheximide (CHX) treatment or *UPF1* knockdown significantly restored the expression of these PTC-containing isoforms (Fig. 4c and d). On the other hand, there was no obvious NMD observed in the remaining six splicing factors (Supplementary Fig. 8), suggesting that these alternatively spliced isoforms are translated into protein variants that might have distinct functions or activities.

To further investigate whether the DAP3-triggered NMD of splicing factors indeed contributes to DAP3-modulated splicing, *RBM6*, a known cancer-related splicing regulator[17] was selected for further investigation. Skipping of *RBM6* exon 6 causes an open reading frameshift and introduces a PTC on exon 8 (Fig. 5a). DAP3 was found to bind to *RBM6* exon 6 and flanking exons and introns (Fig. 5b). Consistent with the observation that *DAP3* depletion promoted NMD of *RBM6* mRNA transcript (Fig. 4c and d), the mRNA level of the canonical exon 6-included *RBM6* isoform was significantly reduced in *DAP3*-KD cells (Fig. 5c). A reduction in protein expression of RBM6 was also confirmed upon *DAP3* depletion (Fig. 5d). By performing RNA-Seq analysis of *RBM6* KD cells, we identified a total of 94 alternative splicing events, which were modulated by RBM6 and DAP3 (defined as "co-modulated" targets) (Fig. 5e and f; Supplementary data 5).

Notably, *DAP3*-KD-induced splicing changes in five validated co-modulated targets could be rescued by the restoration of RBM6 expression (Fig. 5g and h; Supplementary Fig. 9). Altogether, our results indicated that DAP3 could modulate genome-wide changes in splicing indirectly via affecting the expression of splicing factors/regulators.

**Clinical relevance of DAP3-modulated mis-splicing in cancers.** It has been reported that DAP3 is overexpressed in a broad range of cancer types and appears to have strong oncogenic effect[11]. We next conducted a pan-cancer analysis of alternative splicing across 33 cancer types using the TCGA Spliceseq dataset[18] to study whether DAP3-modulated splicing events are frequently dysregulated in different cancer types and evaluate their clinical significance. Of 20 experimentally validated DAP3-modulated splicing events (Figs. 2f, 4b, and 5g), 18 of them (90%) were detected in almost all 33 TCGA cancer types with varying PSI values (Supplementary Fig. 10). To examine whether these DAP3-modulated splicing events are dysregulated in cancers, we compared the PSI values of these splicing events between tumors and non-tumor (NT) samples in several representative cancer types, which demonstrate significantly higher expression of DAP3 in tumors as reported previously[11]. We found that DAP3-modulated splicing events (e.g., *WSB1, SNRPB, TIAL1, TBL1X, SSBP3*, and *CADM1*) were significantly dysregulated in tumors compared to NT samples in multiple cancer types, such as esophageal carcinoma (ESCA), breast invasive carcinoma (BRCA), colon adenocarcinoma (COAD), glioblastoma multiforme (GBM), liver hepatocellular carcinoma (LIHC), and stomach adenocarcinoma (STAD) (Fig. 6a). Moreover, lower PSI values of *WSB1, TBL1X*, and *SNRPB* gene in tumors were significantly correlated with the shorter overall survival (OS) time of patients with ESCA and LIHC, whereas the higher PSI values of *TIAL1* and *CADM1* in tumors predicted poorer prognosis in patients with LIHC and COAD (Fig. 6b). It is known that different cancer types display diverse alternative splicing landscapes[19]. We further conducted a comparative analysis of DAP3-modulated alternative splicing using the RNA-Seq datasets of *DAP3*-depleted ESCC cells and 13 matched pairs of tumors and NT samples from the TCGA ESCA. Same as *DAP3*-depleted ESCC cells, SE is the major type of differentially regulated splicing type in ESCA tumors, followed by MXE, A5SS, A3SS and IR (Figs. 2a; 6c). Among 3811 differentially spliced events (across 1606 genes) detected in tumors (FDR < 0.05, |ΔPSI| ≥ 10%), 14% (535/3,811) of which were identified as "DAP3-modulated splicing events" (Fig. 6c–e; Supplementary data 6). For example, DAP3-modulated splicing of *WSB1, SNRPB, TIAL1*, and *TBL1X* gene were significantly altered in tumors compared to their matched NT samples (Fig. 6f). All these observations

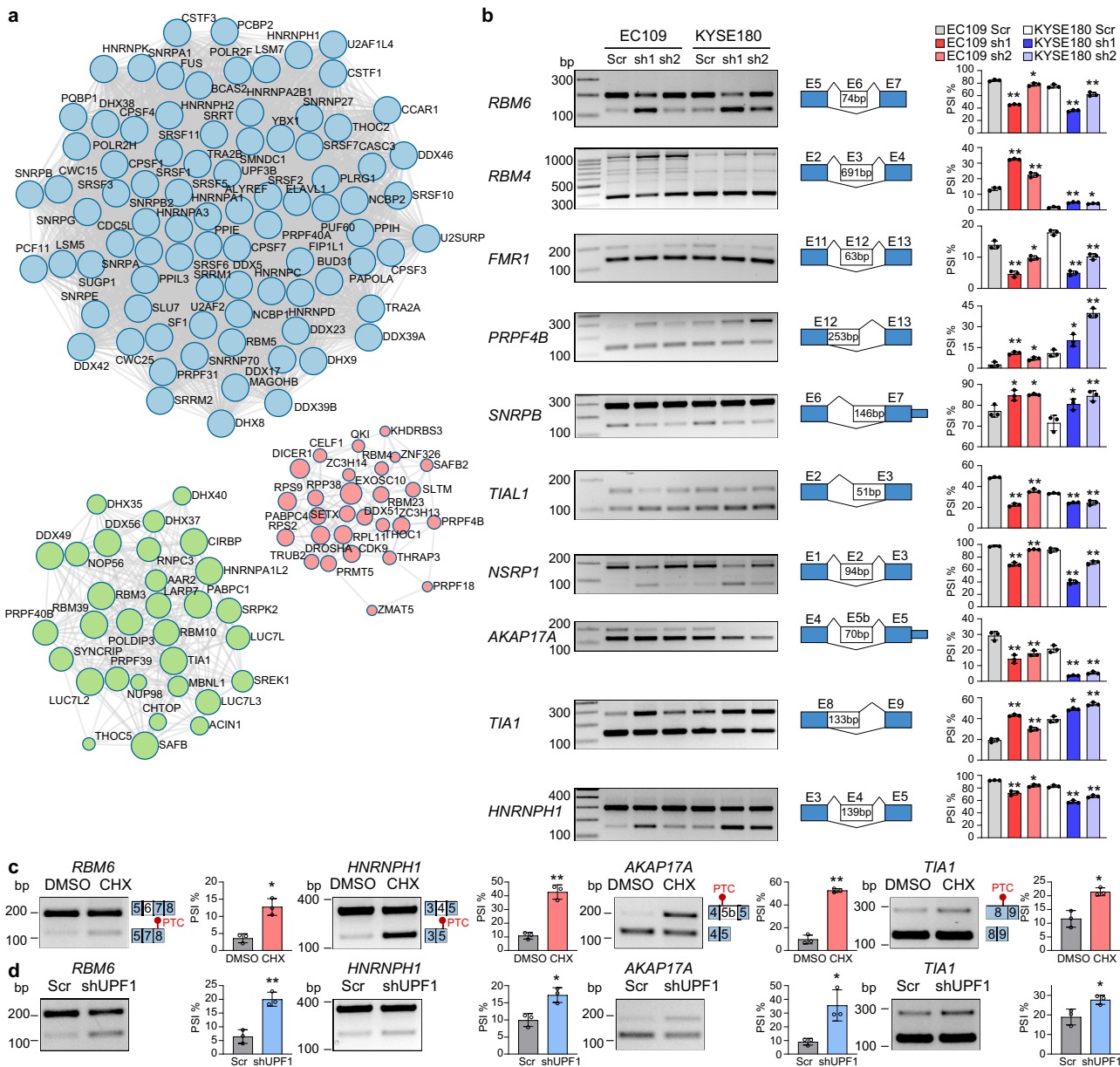

**Fig. 4 DAP3 orchestrates splicing regulatory networks by modulating alternative splicing of splicing factors. a** Functional protein association networks of genes that are involved in mRNA processing and have their splicing modulated by DAP3 were identified in Metascape[45](http://metascape.org). **b** Semiquantitative RT-PCR analyses of splicing changes of 10 representative splicing factors *RBM6, RBM4, FMR1, PRPF4B, SNRPB, TIAL1, NSRP1, AKAP17A, TIA1,* and *HNRNPH1* upon *DAP3* depletion in EC109 and KYSE180 cells. **c, d** Semiquantitative RT-PCR analyses of splicing changes of *RBM6, HNRNPH1, AKAP17A,* and *TIA1* upon inhibition of NMD in EC109 cells. Cells were treated with CHX or vehicle control (DMSO) for 6 h (**c**) or transfected shUPF1 or the scramble control (Scr) for 48 h (**d**). **b–d** Data are represented as mean ± s.d. of *n* = 3 biologically independent samples. Statistical significance is determined by unpaired, two-tailed Student's *t*-test (**p* < 0.05, ***p* < 0.01). Exact p-values and source data are provided in Source Data file.

suggest that prevalent dysregulations of DAP3-modulated splicing events are clinically relevant to multiple cancer types.

**DAP3 increases WSB1 expression via repressing AS-NMD of *WSB1* to promote tumorigenesis.** We next looked for experimental evidence that supports a functional link of DAP3-modulated mis-splicing to tumorigenesis. Here, we were particularly interested in one of the DAP3 target genes, *WSB1* (WD repeat and SOCS box-containing protein 1) which is an E3-ubiquitin ligase that promotes ATM ubiquitination and degradation to drive tumorigenic progression[20], with the following reasons including (1) differentially spliced-in tumors of multiple cancer types (Fig. 6a), (2)

lower PSI value in tumors predicts poor prognosis (Fig. 6b), and (3) hypothetically, its expression can be affected by DAP3 via AS-NMD (Fig. 7a). As shown in Fig. 2f and g, inclusion of a non-canonical exon E6a or E6b by usage of cryptic splicing sites at intron 5 in the *WSB1* mRNA transcript was promoted by *DAP3*-KD or KO. While exon E6b has an additional 12-nucleotide sequence at 3' end than exon E6a, inclusion of either E6a or E6b introduces a PTC that may lead to selective degradation by NMD (Fig. 7a). Suppression of NMD by either CHX treatment or *UPF1* knockdown significantly restored the expression of E6a/E6b-included *WSB1* isoform, confirming that inclusion of exon E6a or E6b truly introduces a PTC-containing isoform and triggers NMD (Fig. 7b). Both KD and KO

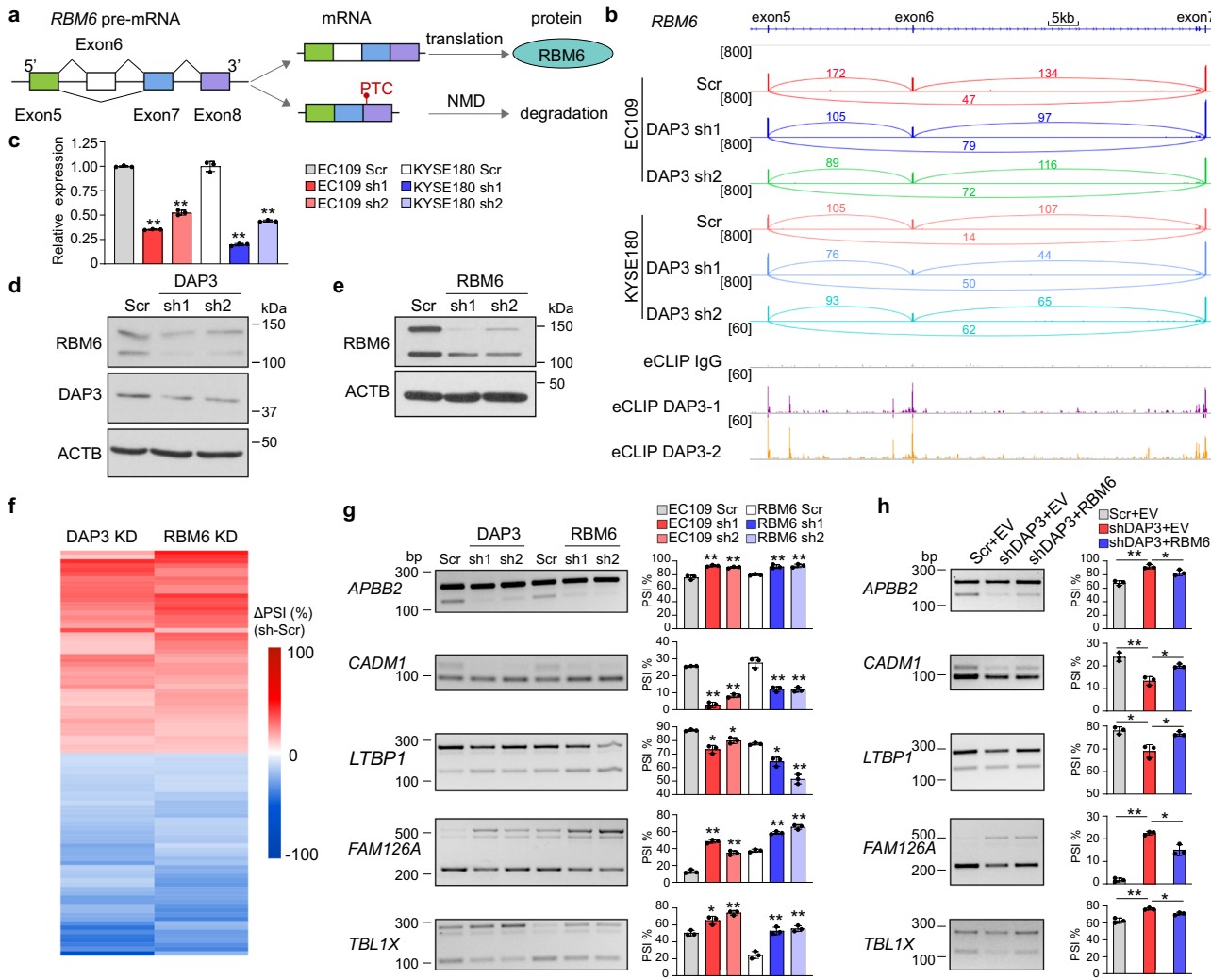

**Fig. 5 DAP3 positively regulates RBM6 expression via repressing AS-NMD of *RBM6*. a** Schematic diagram depicting skipping of *RBM6* exon 6 causes a shift in the open reading frame, resulting in the introduction of a PTC into *RBM6* transcript and possible NMD. **b** Visualization of RNA-Seq data of *DAP3*-depleted EC109 and KYSE180 cells and DAP3 eCLIP-seq peaks spanning the *RBM6* gene locus using IGV. Significant peaks are marked by purple and orange bars. **c** qRT-PCR analysis of expression of the *RBM6* exon 6-included isoform after *DAP3* depletion in EC109 and KYSE180 cells. *ACTB* was used as a housekeeping gene internal control. Data are represented as mean ± s.d. of technical triplicates. **d** Western blot analysis of RBM6 protein expression after *DAP3* depletion in EC109 cells. **e** Western blot analysis of RBM6 protein expression after *RBM6* knockdown in EC109 cells. **f** Heatmap showing the co-modulated splicing events upon knockdown of *DAP3* and *RBM6*. **g** Semiquantitative RT-PCR analyses of five randomly selected splicing events co-modulated by DAP3 and RBM6. Data are represented as mean ± s.d. of *n* = 3 biologically independent samples. **h** Semiquantitative RT-PCR analyses of the indicated splicing events after re-expressing RBM6 in *DAP3*-depleted cells. Data are represented as mean ± s.d. of *n* = 3 biologically independent samples. **c**, **g**, **h** statistical significance is determined by unpaired, two-tailed Student's *t*-test (*p < 0.05, **p < 0.01). Exact p-values and source data are provided in Source Data file.

of *DAP3* significantly decreased the expression of WSB1 at mRNA and protein levels through repressing the skipping of E6a/E6b (Fig. 7c–e). Conversely, overexpression of DAP3 increased WSB1 protein expression (Fig. 7f). Restoring DAP3 expression in KO cells could rescue WSB1 expression (Supplementary Fig. 11a). As a downstream target of WSB1[20], the protein expression of ATM was negatively regulated by DAP3-mediated change in WSB1 protein expression (Fig. 7d–f).

Previously we have shown DAP3-promoted tumorigenesis[11], but whether DAP3-modulated splicing contributes to its oncogenic function is not clear. As shown above, DAP3 depletion caused AS-NMD of *WSB1* to suppress WSB1 expression. Therefore, we hypothesized that DAP3-modulated AS-NMD of *WSB1* might contribute to the oncogenic function of DAP3. To this end, we examined the changes in tumorigenic ability of WT or *DAP3*-KO cells after restoring *WSB1* expression by performing in vitro

foci formation and anchorage independent soft agar assays and in vivo xenograft assay (Fig. 7h–k). Overexpression of *WSB1* in WT EC109 cells promoted tumorigenesis in vitro and in vivo when compared to WT cells expressing empty vector control (Fig. 7h–k), indicating the oncogenic role of *WSB1* in cancer cells. While KO of *DAP3* significantly repressed the tumorigenicity, reintroduction of *WSB1* gene into *DAP3*-KO cells significantly attenuated *DAP3*-KO-induced suppression of tumorigenicity both in vitro and in vivo (Fig. 7h–k). Similarly, restoring WSB1 expression in the *DAP3*-KD cells also partially rescued the reduced tumorigenicity (Supplementary Fig. 11b–e). These data indicate that DAP3 depletion represses tumorigenesis at least partially through promoting AS-NMD of *WSB1* gene. In sum, our functional investigation of nonproductive splicing of *WSB1* supports a causal relationship between DAP3-modulated mis-splicing and tumorigenesis.

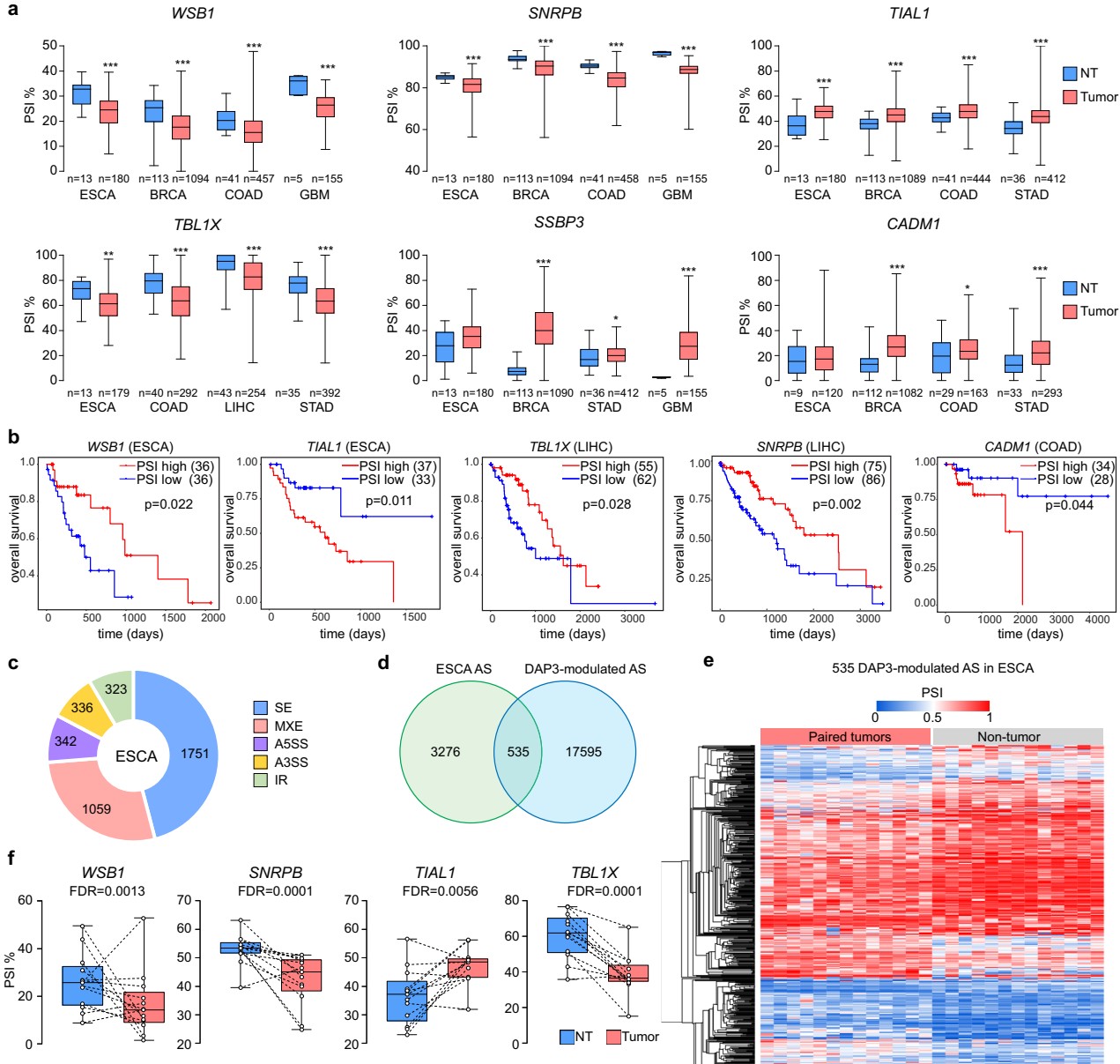

**Fig. 6 Clinical relevance of DAP3-modulated mis-splicing in multiple cancer types. a** Boxplots showing the PSI values of six experimentally validated DAP3-modulated splicing events in tumors and their matched NT samples of six representative TCGA cancer types including esophageal carcinoma (ESCA), breast invasive carcinoma (BRCA), colon adenocarcinoma (COAD), glioblastoma multiforme (GBM), liver hepatocellular carcinoma (LIHC), and stomach adenocarcinoma (STAD). Statistical significance was determined by two-tailed nonparametric Mann–Whitney test (*$p < 0.05$, **$p < 0.01$, ***$p < 0.001$). **b** Kaplan–Meier OS plots comparing patients demonstrating the top 25th percentile of PSI values of the indicated DAP3-modulated splicing events ("PSI high" group) and those with the bottom 25th percentile of PSI values ("PSI low" group). Statistical significance was determined by log-rank test. **c** Pie chart showing the distribution of each type of differentially spliced events in 13 matched pairs of ESCA tumors and their NT samples detected by rMATS analysis[15]. **d** Venn diagram showing the number of significantly altered spliced events identified by RNA-Seq analysis of both *DAP3*-depleted ESCC cells (EC109 and KYSE180) and the TCGA ESCA. **e** Heatmap showing 535 significantly altered DAP3-modulated splicing events detected in both ESCC cell lines and ESCA tumors as described in **d**. The heatmap was generated using hierarchical clustering of tumors and NT samples (average linkage and Euclidean distance used). The color spectrum indicates PSI values. **f** Boxplots showing the PSI values of four representative DAP3-modulated splicing events in 13 matched pairs of ESCA tumors and NT tissues. **a, f** The box extends from the 25th to 75th percentiles. The line in the middle of the box is plotted at the median. The whiskers indicate min to max. Exact *p*-values and source data are provided in Source Data file.

## Discussion

It is known that high level of fidelity required for splicing needs additional action of a complex interplay of RBPs that bind adjacent to splicing sites and promote recruitment of the spliceosome or outcompete spliceosomal components for binding to target RNAs[21]. Even modest changes in the abundance or activity of individual RBPs or core spliceosomal proteins can result in

aberrations or mistakes in splicing, which may be deleterious to cells and may result in cell death or cellular transformation[8,22]. In this study, through application of eCLIP-Seq, RNA-Seq, and proteomics analyses, we characterize DAP3 as a widespread alternative splicing regulatory RBP which modulates thousands of splicing events and dissect its associated regulatory mechanisms and functional relevance to cancer. DAP3, which is overexpressed

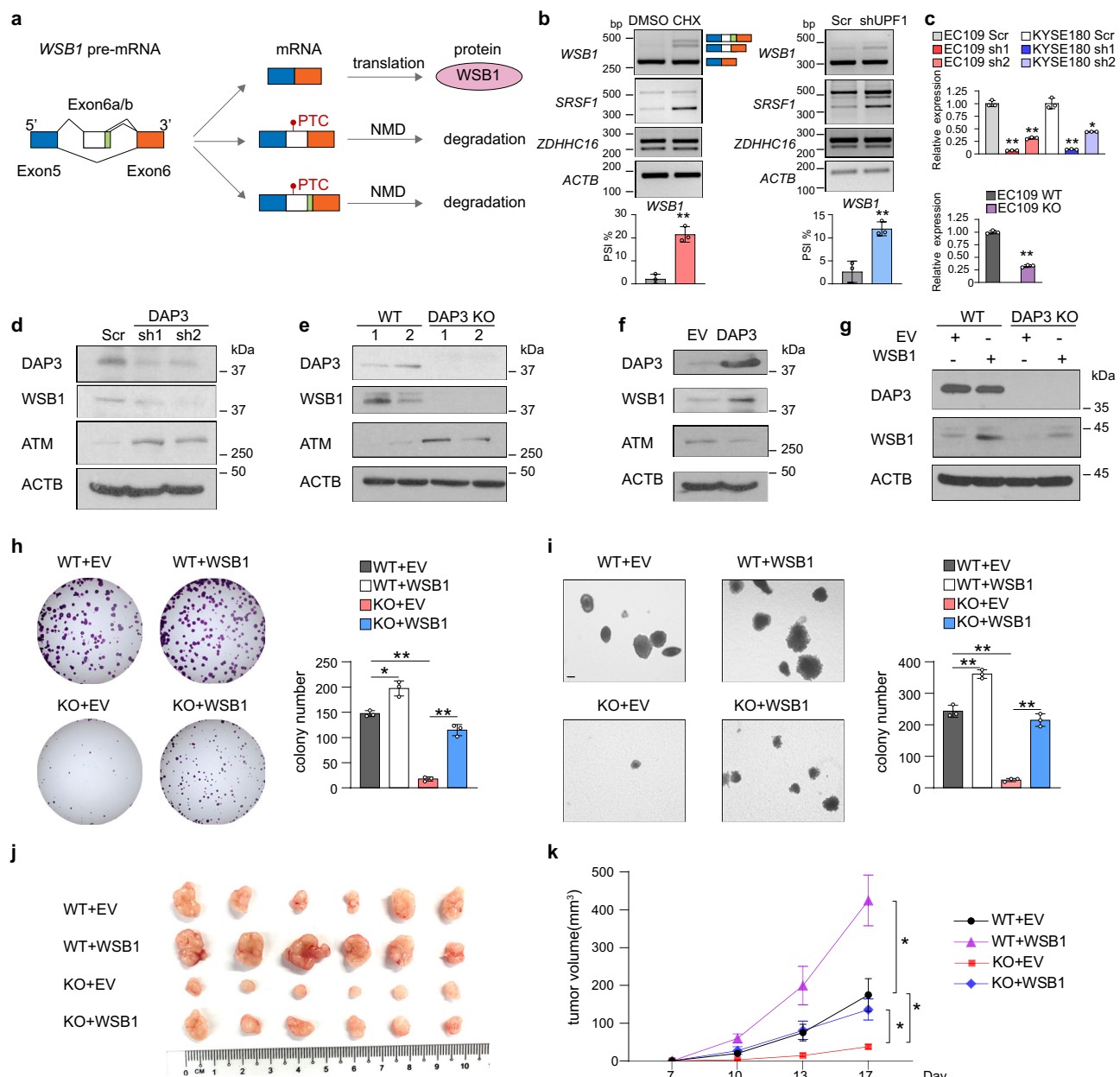

**Fig. 7 DAP3 increases WSB1 expression via suppressing AS-NMD of *WSB1* to promote tumorigenesis. a** Schematic diagram illustrating the inclusion of non-canonical exon E6a and E6b introduces a PTC to *WSB1* mRNA transcript, which may result in NMD. **b** Semiquantitative RT-PCR analysis of the *WSB1* splicing upon inhibition of NMD. Left panel: EC109 cells were treated with CHX or vehicle control (DMSO) for 6 h. Right panel: EC109 cells were transfected with shUPF1 or the scramble control (Scr) for 48 h. *SRSF1* or *ZDHHC16* serves as a positive or negative control, respectively, to ensure successful inhibition of NMD. *ACTB* was used as a housekeeping gene internal control. Data are represented as mean ± s.d. of n = 3 biologically independent samples. **c** qRT-PCR analysis of the change in expression of *WSB1* canonical isoform (E6a/6b-skipped) upon *DAP3*-KD (upper panel) or KO (lower panel). *ACTB* was used as a housekeeping gene internal control. Data are represented as mean ± s.d. of technical triplicates. **d**–**f** Western blot analyses of DAP3, WSB1, ATM, and ACTB protein expression in **d** *DAP3*-KD EC109 cells, **e** *DAP3*-KO EC109 cells ("1" and "2" indicate two different WT or KO clones), and **f** EC109 cells with DAP3 overexpression. EV empty vector control. **g** Western blot analysis of DAP3 and WSB1 protein expression in *DAP3*-KO or WT EC109 cells that were overexpressed with the empty vector control or *WSB1* construct. **h, i** Quantification of foci formation (**h**) or soft agar colony formation (**i**) induced by the indicated stable cells. Scale bar: 200 μm. Data are represented as mean ± s.d. of n = 3 biologically independent wells. **b**, **c**, **h**, **i** Statistical significance is determined by unpaired, two-tailed Student's t-test (*p < 0.05, **p < 0.01). **j** Xenograft tumors derived from the indicated stable cell lines at end point (n = 6 mice per group). **k** Growth curve of tumors derived from the indicated cells in mice, over a 17-day observation period. Data are presented as the mean ± s.e.m. Statistical significance is determined by unpaired, two-tailed Student's t-test (*p < 0.05). Exact p-values and source data are provided in Source Data file.

in multiple cancer types, has been proven to be a cancer-promoting gene[11]. DAP3 has been reported to interact with ADAR proteins and repress A-to-I RNA editing; however, it does not bind proximal to the editing site within target RNA[11]. Here, our eCLIP-Seq analysis revealed a widespread binding of DAP3 to RNAs transcribed from 9699 genes (approximately one-third of the total number of genes in the human genome). Many of these bound genes are critical regulators of mRNA processing including splicing and polyadenylation, gene expression, and mitotic cell cycle regulation. Of note, we found that DAP3 has an RNA-binding preference for exonic sequences with two identified top enriched binding motifs that are splicing regulatory sequence motifs, GAAGAA and AGGUAAGU. The purine-rich GAAGAA hexamer, appearing to be one of the strongest ESEs critical for constitutive and alternative splicing[13], has also been found as an internal exonic binding motif of SR proteins, such as SRSF1[23,24], transformer 2 alpha homolog (TRA2A), and transformer 2 beta homolog (TRA2B)[25]. The RNA-binding preference of DAP3 towards the GAAGAA hexamer may partially explain why DAP3 tends to promote exon inclusion in both cancer cell lines. The other motif AGGUAAGU is a consensus sequence at the 5' splice site, which is complementary to the nucleotides 4–11 of U1RNA[14]. In concordance with this finding, we observed that DAP3 could modulate approximately a thousand A5SS events with a slight preference for the proximal 5' splice site. However, such regulations may also be controlled via other *cis-* and/or *trans*-acting mechanisms such as the strength of 5' splice sites and/or involvement of other splicing factors/regulators, which bind to the 5' splice sites.

Although we found a significant enrichment of DAP3-binding peaks in genes undergoing DAP3-modulated splicing and provided experimental evidence supporting the binding of DAP3 to its target gene *WSB1* is required for DAP3-mediated splicing change in *WSB1*, it is unlikely that extensive splicing changes mediated by DAP3 are simply through its binding to splicing consensus sequence or regulatory elements. Moreover, we also showed that the binding affinity of DAP3 to RNA targets is not associated with the strength of its splicing regulation. In this study, we delineated two distinct mechanisms of how DAP3 functions in splicing regulation (Fig. 8). First, DAP3 directly binds to target RNA and mediates the recruitment of splicing factors such as SFPQ and NONO to the binding sites via protein–protein interaction independent of their interactions with RNA. SFPQ and NONO belong to the multifunctional Drosophila behavior/human splicing (DBHS) family of proteins with highly conserved N-terminal RNA recognition motifs (RRMs). Although they are not essential components for spliceosome assembly, numerous studies have identified them as spliceosome-associated proteins and play an important role in alternative splicing[26–30]. Using a proteomic approach, we identified SFPQ and NONO as DAP3 interactors. A previous study has demonstrated a SFPQ interacting RBP, Dido3, is required to recruit SFPQ to its target exon for efficient alternative splicing[31]. Similarly, we observed that loss of DAP3 could significantly compromise the association of SFPQ and NONO with their target RNA transcripts, indicating DAP3 functions as an important mediator to facilitate binding of SFPQ and NONO to RNA. Overexpression of SFPQ and NONO could readily rescue splicing changes of several exemplary DAP3 target transcripts *WSB1, SSBP3,* and *RHBDD2* in *DAP3*-depleted cells. Another mechanism of DAP3-modulated splicing involves indirect modulation of splicing by fine-tuning the splicing pattern of hundreds of splicing factors, as supported by our eCLIP-Seq and RNA-Seq data. These splicing factors include key components of splicing machinery, such as spliceosomal small nuclear ribonucleoproteins and heterogeneous nuclear ribonucleoproteins, as well as important splicing regulators, such as SR splicing factors, RBM proteins, and DEAD-box helicases. These splicing factors form dynamic splicing regulatory networks, which control splicing of tens of thousands of target genes. DAP3 alters the expression of these splicing factors via nonproductive splicing or isoform switching, thereby coordinately modulating splicing of multiple genes via their respective splicing factors. This is exemplified by DAP3-mediated AS-NMD of *RBM6* gene in this study. Depletion of DAP3 induces skipping of *RBM6* exon 6, causing a frameshift in its coding sequence and triggers NMD, indicating a positive

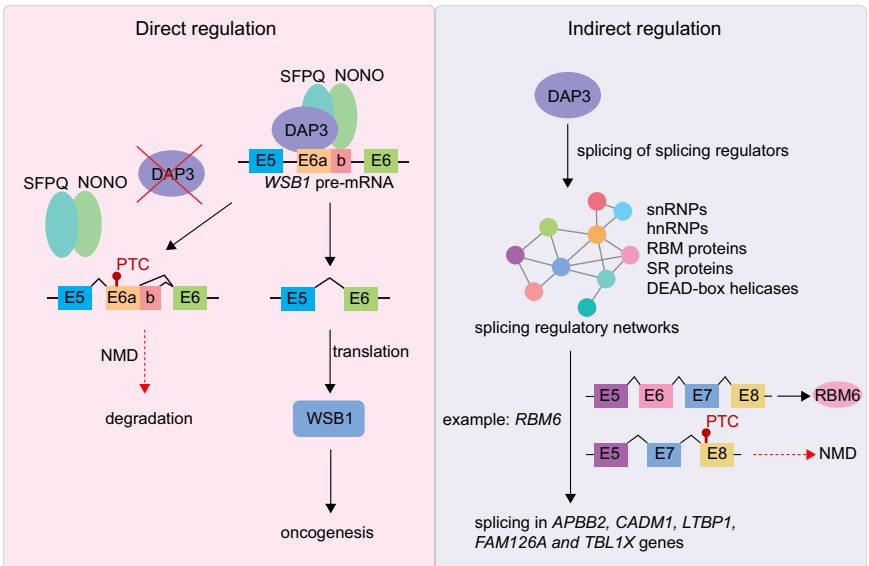

**Fig. 8 Multilayered control of splicing regulatory networks by DAP3 leads to widespread alternative splicing changes in cancer.** There are two distinct mechanisms of how DAP3 exerts its splicing regulatory functions. First, DAP3 directly binds to target RNA and mediates the recruitment of splicing factors, such as SFPQ and NONO, to the binding sites. Another mechanism involves indirect modulation of splicing through fine-tuning the splicing pattern of hundreds of splicing factors by DAP3. When DAP3 is overexpressed in many cancer types, global splicing changes can be observed and contribute to tumorigenesis.

regulation of DAP3 on RMB6 expression. Similar to DAP3, RBM6 also preferentially binds to exonic sequences; however, it has a distinct binding preference for sequences with CUCUGAA motif[17]. DAP3 cross-regulates RBM6, which in turn leads to a cascade of secondary splicing changes, expanding the repertoire of DAP3-affected splicing events. However, we also observed approximately a hundred splicing events that are discordant in DAP3-depleted and RBM6-depleted cells. Because splicing regulation often involves numerous RBPs and we showed DAP3-modulated splicing of hundreds of RBPs, it is possible that the modulation of these discordant splicing events involves other DAP3-modulated RBPs other than RBM6 alone. Therefore, the overall effects of these discordant splicing events after DAP3 depletion were not dominated by RBM6 downregulation.

The precise coordination and regulation of RNA processing by RBPs is essential for maintaining cellular homeostasis, which may otherwise result in various diseases including cancer. Our previous study demonstrated that DAP3 is overexpressed in multiple cancer types[11]. In this study, we demonstrate that many DAP3-modulated splicing events were significantly altered in patients' tumors and such changes demonstrate prognostic values in multiple cancer types. Here we conducted detailed functional analyses of the AS-NMD of WSB1 gene, whose alternative splicing could account for the oncogenic phenotypic changes driven by DAP3. WSB1 contains seven WD40 repeats and a SOCS box at the C-terminus and functions as an E3-ubiquitin ligase in ubiquitination and proteasomal degradation[32]. It regulates the metastatic potential of renal carcinoma, osteosarcoma, and hormone receptor negative breast cancer by modulating pVHL, RhoGDI2, metalloproteinase (MMP) activity, vascular endothelial growth factor (VEGF) secretion[33–35]. More importantly, it ubiquitinates ATM for degradation to overcome oncogene-induced senescence, thereby playing an important role in tumor initiation[20]. Besides transcriptional regulation by oncogenic transcription factors, such as c-Myc, HIF-1, and CREB-ATF[36,37], our study provides another mechanism of WSB1 expression regulation in cancer cells, which is via DAP3-modulated alternative splicing. DAP3 suppresses the nonproductive splicing of WSB1, which is prone to NMD. Since DAP3 is widely overexpressed in cancer cells, such DAP3-driven stabilization of WSB1 mRNA transcripts may be a key step in tumor initiation.

In sum, our findings demonstrate that DAP3 coordinates splicing regulatory networks to modulate global alternative splicing in cancer via both RNA–protein and protein–protein interactions. Targeting DAP3-driven splicing events and blocking the splicing regulatory ability of DAP3 and/or specifically targeting DAP3-driven splicing events may hold great promise for cancer treatment.

## Methods

**Cell culture**. EC109 and KYSE180 cells were cultured in HyClone RPMI 1640 medium (Thermo Fisher Scientific) supplemented with 10% FBS at 37 °C in a humidified incubator containing with 5% CO2. HEK293T cells were cultured in DMEM high glucose medium (Biowest) supplemented with 10% FBS at 37 °C in a humidified incubator containing with 5% CO2.

**Generation of stable knockdown (KD) and overexpression cells**. The DAP3-KD stable EC109 and KYSE180 cell lines were established using lentiviral transduction, followed by puromycin selection. The pLKO-DAP3-sh1 (5'GCTTATCCAGCTATAC GATAT3'), pLKO-DAP3-sh2 (5'ATCCTGGTTTCCAACTATAAC3'), pLKO-RBM6-sh1 (5'GACTGGTCTTCAGATACAAAT3') and pLKO-RBM6-sh2 (5'ACGGAACAC AAGTAGACTTTA3') constructs were used for the lentivirus packaging. EC109 cells stably expressing Flag-tagged WSB1 were established by transduction of packaged lentiviral CSII-CMV-Flag-WSB1 constructs, followed by the puromycin selection.

**Generation of DAP3-knockout (KO) cells using CRISPR/Cas9 system**. DAP3 sgRNAs were designed using the CRISPR design tool from The Massachusetts Institute of Technology (MIT) (http://crispr.mit.edu). The DAP3 sgRNA

(ATAGCTCTCGGACTCTCAAC) targeting exon 3 of DAP3 was cloned into the pX330A vector. EC109 cells were transfected with either empty vector or vector expressing DAP3 sgRNA and split into single cell. Clones grew from single cell were lysed by DirectPCR Lysis Reagent (Viagen Biotech) followed by the detection of indels in each single clone by T7EI assays. PCR products were TA cloned and Sanger sequenced to confirm the biallelic KO of DAP3 in each individual clone. Western blot analysis was performed to confirm DAP3-KO at protein level.

**eCLIP-Seq data analysis**. The eCLIP experiment was performed as previously described[38]. Briefly, 20 million of EC109 cells were UV crosslinked, fragmented and immunoprecipitated using a DAP3 antibody (Abcam, ab2637) or control IgG (Invitrogen, 02-6202). Next, the bound protein-RNA products were subjected to gel electrophoresis and membrane transfer. Bound RNAs on the membrane corresponding to the protein size of DAP3 and 70 kDa above were extracted and further processed with adaptor ligation. The cDNA library was prepared by revere transcription and sequenced by paired-end 100 bp sequencing performed on the Illumina HiSeq 4000 platform. The sequencing data were processed as previously described and clusters identified in IP samples were compared against paired size-matched input to obtain significantly enriched peaks using a Fisher's Exact test (or Yates' Chi-Square test if all observed and expected values were above 5), with p-values reported not corrected for multiple hypothesis testing[7,38]. Peaks with fold enrichment (four fold) and significance (p-value <0.001) in immunoprecipitation versus paired size-matched input sample were defined as significant binding peaks. The eCLIP-seq data were first utilized in our previous study[11] (GEO accession number: GSE144318); however, the comprehensive analysis of eCLIP-seq data were provided only in the present study. The gene ontology pathway enrichment analysis was done using WebGestalt[39] portal (http://www.webgestalt.org/). The significance of enrichment for GO sets were evaluated by the pipeline default hypergeometric test.

**Coverage plots of eCLIP peaks across transcripts and splice junctions**. To understand the distribution of DAP3-binding peaks on transcripts, a previous method was adapted to generate the coverage plot for the eCLIP-Seq peaks[40]. To increase the reliability of this analysis, we only considered "expressed genes" which have TPM (transcripts per million) ≥ 1 in each of 10 RNA-Seq datasets of EC109 cells generated by our team previously, resulting in 8170 such genes. The gene annotations for reference human genome (hg19) were downloaded from UCSC Table Browser[41] ("knownGene" table), and only the longest isoform was considered as the representative transcript per gene. Next, each gene was split into 183 bins excluding the introns (13 for 5'UTR, 100 for CDS, and 70 for 3'UTR) proportional to the median 5'UTR, CDS, and 3'UTR lengths of the representative transcripts of expressed genes in EC109 cells. Finally, by using the significant peaks from the DAP3 eCLIP-Seq output, the cumulative peak coverage for each bin was calculated for these genes in a strand-specific manner with "bedtools coverage"[42] and used to generate the final plot.

Similarly, the coverage plots for splice junctions were generated as described above. For each type of splicing events (SE, A5SS, A3SS, MXE, IR) identified by rMATS, the schematic representation of exons, introns and splice junctions involving the AS event is depicted as Fig. 2d. Based on the above schematic, each of the exonic and intronic regions were split into 100 bins, and the coverage of significant AS events in each bin was calculated in a strand-specific manner by using "bedtools coverage". In case multiple events of the same type were present in the same gene, each was regarded as a different event as long as the upstream or downstream boundaries were different, otherwise the longest inclusive form of the alternatively spliced exon was considered. The cumulative coverage of the significant peaks from DAP3 eCLIP-Seq output was then plotted for each type of AS events.

**Differential gene expression and alternative splicing analysis of RNA-Seq data**. Gene expression profiling and differential expression analysis were performed by using CSI NGS Portal[43] (https://csibioinfo.nus.edu.sg/csingsportal). Briefly, strand-specific RNA-Seq was performed as previously described[11]. Briefly, clean reads were aligned to the reference human genome (hg19) by using STAR[44] with default parameters. The differential alternative splicing (AS) events between DAP3-KD duplicates and the scrambled control were identified by using rMATS[15] for five major types (SE, A5SS, A3SS, MXE, IR). The FDR was calculated using the default parameters based on the Benjamini–Hochberg approach. In the rMATS output, only significant AS events were further considered defined as: sum(IJC_KD1, SJC_KD1) ≥ 20; sum(IJC_KD2, SJC_KD2) ≥ 20; sum(IJC_Scr, SJC_Scr) ≥ 20; abs(IncLevelDifference) ≥ 0.1; FDR < 0.05. The GO pathway enrichment analysis of DAP3-modulated alternatively spliced gene was done using WebGestalt[39] portal (http://www.webgestalt.org/). The protein–protein interaction network analysis of DAP3-modulated alternatively spliced RBPs was done using Metascape[45] portal (http://metascape.org).

RNA-Seq datasets of 13 matched pairs of ESCA tumor and NT tissues from the TCGA were used for transcriptome-wide splicing analysis using the same rMATS pipeline. Differentially spliced events in tumors relative to NT samples were identified based on the following filter criteria: |ΔPSI (tumor vs normal)| ≥ 0.1 and FDR < 0.05. The heatmap of differentially spliced events was generated using

hierarchical clustering (average linkage and Euclidean distance used) on Morpheus (https://software.broadinstitute.org/morpheus).

*Motif enrichment analysis.* Motif enrichment analyses of DAP3 eCLIP peak sequences, sequences of DAP3-modulated cassette exons and sequences of DAP3 eCLIP peaks located within the region from the upstream to downstream constitutive exon of DAP3-modulated splicing event were performed using "findMotifsGenome.pl" function in HOMER[12] for "de novo" motif enrichment with the flag "-rna". The background is randomly selected sequences from the reference genome, corrected for sequence content (GC content and other bi-nucleotide composition, http://homer.ucsd.edu/homer/motif/ and http://homer.ucsd.edu/homer/motif/rnaMotifs.html for details). Statistics was derived from binomial test against the random genomic background using the HOMER default setting.

*TCGA SpliceSeq analysis of DAP3-modulated splicing events.* Data from the TCGA SpliceSeq analysis[18] of 33 cancer types were downloaded from http://bioinformatics.mdanderson.org/TCGASpliceSeq. Out of the 20 validated DAP3-modulated splicing events, 18 were manually mapped into the splicing events identified in TCGA SpliceSeq dataset according to the alternative splice site locations. The PSI values of each splicing event in tumor and NT samples were compared and statistical significance was determined by two-sided nonparametric Mann–Whitney test. For the OS analysis, we segregated the TCGA patients based on the PSI values of each DAP3-modulated splicing event. The patients with the top 25th percentile of PSI values were defined as "PSI high" group, while patients demonstrating the bottom 25th percentile of PSI as the "PSI low" group. The OS benefit was estimated by the log-rank test and presented by a Kaplan–Meier plot.

**RNA purification and semiquantitative RT-PCR.** Total RNA was purified using RNeasy Mini Kit (Qiagen) with RNAse-Free DNase Set (Qiagen) to digest contaminating DNA. Reverse transcription was performed using SensiFAST™ cDNA synthesis kit (Bioline) and PCR was performed using exTEN 2× PCR Master Mix (Axil Scientific). RT-PCR products were subjected to electrophoresis in 2–3% agarose gel with ethidium bromide and visualized. Molecular weight markers (unit: bp) are labelled in gel images. Band intensity were measured by ImageJ for PSI calculation. Primer sequences are listed in Supplementary data 7.

*Splicing minigene assay.* The genomic DNA fragment spanning from exon 5 to exon 6 of WSB1 gene was cloned into pcDNA3.1(+) plasmid and deletion mutants were generated by mutagenesis cloning using PrimeSTAR Max DNA Polymerase (Takara). Minigene plasmids were transfected into cells using lipofectamine 2000. After 48 h total RNA was purified using RNeasy Mini Kit (Qiagen) with RNAse-Free DNase Set (Qiagen) to digest contaminating DNA. Reverse transcription was performed using SensiFAST™ cDNA synthesis kit (Bioline) and PCR was performed using pcDNA3.1(+) forward and reverse primers by exTEN 2× PCR Master Mix (Axil Scientific). Primer sequences are listed in Supplementary data 7.

*RNA pulldown assay.* RNA probe was generated by RiboMAX™ Large Scale RNA Production Systems (Promega) using DNA template containing 5′-T7 promoter, WSB1 exon E6b wt or deletion mutant sequences, and 3′-aptamer. A total of 10 μg RNA probe was incubated with 20 μl Dynabeads MyOne C1 (Invitrogen) in 300 μl binding buffer (100 mM NaCl, 10 mM MgCl2, 50 mM Hepes, pH 7.4, and 0.5% Igegal CA-630) for 30 min at 4 °C with rotation followed by three washes with washing buffer (250 mM NaCl, 10 mM MgCl2, 50 mM Hepes, pH 7.4, and 0.5% Igegal CA-630). For each reaction, 1 mg whole-cell extract was diluted in 300 μl washing buffer and supplemented with 2 μl 10 mg/ml yeast tRNA (Invitrogen) and SUPERase•In™ RNase Inhibitor (Invitrogen). RNA immobilized beads were incubated with protein mixtures for 30 min at 4 °C with rotation. After three washes, bound proteins were eluted in 2× Laemmli buffer (Sigma) at 95 °C and analyzed by western blot. Primer sequences are listed in Supplementary data 7.

**Co-immunoprecipitation (Co-IP).** For the pulldown of DAP3, SFPQ and NONO protein, EC109 cells were lysed with prechilled lysis buffer (50 mM Tris-HCl, pH 7.5; 150 mM NaCl; 1% Nonidet P40; 0.5% sodium deoxycholate; 1× EDTA-free cOmplete protease inhibitor (Roche)). The lysates were precleared with Dynabeads™ protein G (Invitrogen) at 4 °C overnight. The precleared lysates were incubated with anti-DAP3 (abcam, ab2637), anti-SFPQ (Santa Cruz, sc-271796) and anti-NONO (Santa Cruz, sc-166702) antibodies for 4 h at 4 °C and subsequently with Dynabeads™ protein G at 4 °C overnight. The Dynabeads™ protein G (Invitrogen) with bound proteins were washed with 150 mM NaCl with 1× EDTA-free cOmplete protease inhibitor for six times and boiled with 2× protein loading buffer for 10 mins at 95 °C to elute bound proteins. Western blot analysis was performed to detect co-IP products. For the RNase A treatment prior to the immunoprecipitation, the total lysates were incubated with 0.1 μg/μl RNase A (Thermo Fisher Scientific) at 37 °C for 10 min.

**Western blot analysis.** Protein lysates were denatured and separated on SDS-PAGE gels, transferred onto polyvinylidene difluoride membranes, and immunoblotted with a primary antibody at 4 °C overnight, followed by incubation with a secondary antibody at room temperature for 1 h. The following antibodies are used

in this study: anti-DAP3 (1:1000, Abcam, ab2637), anti-β-actin (1:5000, Santa Cruz, sc-47778), anti-SFPQ (1:1000, Santa Cruz, sc-271796) and anti-NONO (1:1000, Santa Cruz, sc-166702), anti-WSB1 (1:1000, Novus, NBP2-82049), anti-ATM (1:1000, Santa Cruz, sc-377293), and anti-RBM6 (1:1000, Santa Cruz, sc-376201). β-Actin (ACTB) was used as a loading control. Molecular weight markers (unit: kDa) are labelled in blots.

**Immunoprecipitation coupled to mass spectrometry (IP-MS) in combination with SILAC (stable isotope labelling by amino acids in cell culture).** The DAP3-KO EC109 cells were cultured in "light" (84 mg/ml Arg-0 and 146 mg/ml Lys-0) media, and the WT cells were culture in "heavy" (84 mg/ml Arg-10 and 146 mg/ml Lys-8) media in the "forward" experiment for five consecutive passages. In the "reverse" experiment, the light and heavy labels were swapped. SILAC incorporation rates were >98% in both WT and KO samples. IP was performed using anti-DAP3. In the "forward" or "reverse" experiment, the IP products of the WT-heavy and KO-light or WT-light and KO-heavy samples were mixed at 1:1 ratio, respectively. The MS data acquisition and analysis were performed as described previously[46]. The MS data are provided in Supplementary data 4.

**RNA electrophoretic mobility shift assay (REMSA).** The RNA probes were generated by in vitro transcription using RiboMAX™ Large Scale RNA Production Systems (Promega) and biotin-labelled using Biotin 3' End DNA Labeling Kit (Thermo Fisher Scientific). Recombinant protein was prepared as described[16]. REMSA was performed using LightShift® Chemiluminescent RNA EMSA Kit, according to the manufacturer's protocol (Thermo Fisher Scientific). Briefly, biotinylated RNA probes were heated for 5 min at 80 °C and placed on ice immediately to release secondary structure. The biotinylated RNA probes (0.5 pmol) were incubated with 40 ng of Flag-DAP3 proteins in the binding buffer containing 10 mM HEPES (pH 7.3), 20 mM KCl, 1 mM MgCl2, 1 mM DTT, 100 ng/μl tRNA, and 0.2U/μL SUPERase·In™ RNase Inhibitor (Invitrogen) at room temperature for 30 min. In the RNA competition assay, a 100-fold molar excess of unlabelled RNA probes were preincubated with the reaction mixture at room temperature for 5 min before adding biotinylated probes. Samples were subjected to electrophoresis on a 5% native acrylamide gel, transferred to Amersham Hybond-NX (GEHealthcare) membrane, and detected by chemiluminescence. The probe sequences (motifs highlighted in red) are provided in Supplementary data 7.

**RNA immunoprecipitation (RIP)-quantitative PCR (qPCR) analysis.** Cells were lysed by prechilled lysis buffer (50 mM Tris (pH 7.5), 150 mM NaCl, 1 mM EDTA, 1% Triton, 1×complete protease inhibitor (Roche), and 0.1U/μl SUPERase·In™ RNase Inhibitor (Invitrogen)) and incubated overnight with M2 magnetic beads at 4ºC. Then the M2 magnetic beads were washed six times with TBS buffer (50 mM Tris (pH 7.5), 150 mM NaCl, and 0.02U/μl SUPERase·In™ RNase Inhibitor (Invitrogen)). Bound proteins were eluted with 2× protein loading buffer after boiling at 95 °C for 10 min. Western blot was performed to examine pulldown efficiency. RNAs bound to the M2 magnetic beads were eluted with buffer RLT and purified with RNeasy Mini Kit (Qiagen). cDNA was synthesized using SensiFAST™ cDNA synthesis kit (Bioline) and qPCR was performed using GoTaq® qPCR Master Mix (Promega). Enrichment of pulled down RNAs were normalized to the input RNA expression levels. Primer sequences are listed in Supplementary data 7.

**Foci and soft agar colony formation assay.** For foci formation assay, EC109 cells were seeded at a density of $1 \times 10^3$ per well in six-well plates. Medium was replaced every 3 days. Visible colonies in each well were stained with crystal violet solution (0.1% crystal violet; 25% methanol) and quantified. A representative image of a stained well for each treatment was shown.

For soft agar assay, EC109 ($1 \times 10^3$ per well) cells resuspended in 0.4% low-melting agarose were seeded on top of 0.6% low-melting agarose in six-well plates and incubated for 2 weeks. Visible colonies were stained with crystal violet solution (0.005% crystal violet; 25% methanol) and quantified. A representative image viewed under microscope for each treatment was shown.

**In vivo tumorigenicity assay.** NOD scid gamma (NSG) mice (The Jackson Laboratory, RRID:IMSR_JAX:005557) were maintained in pathogen-free (SPF) facility in NUS Comparative Medicine Department. Less than five mice with same sex were housed in a cage at 20–25 °C and 50% humidity with a 12 h light/dark cycle. For in vivo tumorigenicity assay, $0.5 \times 10^6$ EC109 cells were subcutaneously injected into the left or right flank of 4- to 6-week-old NOD scid gamma (NSG) mice (Fig. 7j): $n = 3$ males and $n = 3$ females for each group; Supplementary Fig. 11e: $n = 2$ males and $n = 3$ females for each group). Tumor growth was monitored, and tumor length (L) and width (W) measured at indicated time points. Tumor volume was calculated by the formula $V = 0.5 \times L \times W \times W$. All animal experiments were approved by and performed in accordance with the Institutional Animal Care and Use Committees (IACUC) of National University of Singapore. All tumors were harvested before or on the day of reaching the IACUC approved tumor size (15 mm at the largest diameter).

**Statistics and reproducibility**. Bioinformatic statistics used default settings for each individual pipeline and portal. All other statistical analyses were performed using Graphpad Prism v9.2.0 or Microsoft Excel 2019. All tests used in the study were two-sided and the exact p-values are provided in source data file. The number of replicates were provided in the figures and legends. Western blot and REMSA experiments were performed at least twice with similar results and representative data are shown.

**Reporting summary**. Further information on research design is available in the Nature Research Reporting Summary linked to this article.

## Data availability

The datasets generated and used in this study are available in the GEO repository, Accession ID: GSE123020 (RNA-Seq), GSE172078 (RNA-Seq), and GSE144318 (eCLIP-Seq). Databank URL: http://www.ncbi.nlm.nih.gov/geo/. All the other data are available within the article and its Supplementary Information. Source data are provided with this paper.

## Code availability

The bioinformatics pipelines for RNA-Seq, eCLIP-Seq, and rMATS are available online at the CSI NGS Portal[43] (https://csibioinfo.nus.edu.sg/csingsportal). Bioinformatics code for downstream analysis is available upon reasonable request.

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

## Acknowledgements

This project was supported by National Research Foundation Singapore; Singapore Ministry of Education under its Research Centres of Excellence initiative; Singapore Ministry of Education's Tier 2 Grants (MOE2018-T2-1-005 and MOE2019-T2-2-008 to L.C.); Singapore Ministry of Education's Tier 3 Grants (MOE2014-T3-1-006 to L.C.); Singapore Ministry of Health's National Medical Research Council (NMRC) Clinician Scientist-Individual Research Grant (CS-IRG, project ID: MOH-000214 to L.C.); NMRC

Open Fund Young Individual Research Grant (OFYIRG19nov-0008:MOH-000537 to
J.H.). We thank and acknowledge Prof Gene W. Yeo (Department of Physiology,
National University of Singapore, Singapore; Department of Cellular and Molecular
Medicine, University of California San Diego, USA) for providing technical supports to
the eCLIP-Seq experiment.

## Author contributions

J.H. performed the experiments and wrote the manuscript. O.A., X.R., and H.Y. conducted the bioinformatics analyses. Y.S., S.J.T., H.S., X.K., V.H.E.N., and D.J.T.T. provided insightful suggestions and experimental materials. H.Q.T. contributed to the eCLIP-Seq experiments. D.K. performed and analyzed SILAC-IP-MS experiments. L.C. supervised the study and edited the manuscript.

## Competing interests

The authors declare no competing interests.
