## [Peer Review File · Nature Communications]

Multilayered control of splicing regulatory networks by DAP3 leads to widespread alternative splicing changes in cancerREVIEWER COMMENTS

Reviewer #1 (Remarks to the Author):

This manuscript by Han et al. describes a deep genomic and molecular exploration of the regulatory networks downstream of DAP3, combining target identification by eCLIP and re-analysis of knockdown/RNA-seq to uncover the role of DAP3 in broadly regulating alternative splicing. I think the results here are quite intriguing, and suggest a significant and unappreciated importance of DAP3 in splicing regulation, with links to cancer-relevant altered splicing programs.

My main area of concern regards the extremely high number of targets (9,699 genes) seen for DAP3, and what this means for the interpretation of some of the resulting analyses. The authors note in the discussion that this is ~1/3 of total human genes, but that includes many tissue-specific and other poorly expressed genes. Indeed the authors note in the methods for eCLIP analysis that to define expressed genes for use in various analyses, "the expressed genes were defined as having TPM ≥ 1 in each of 10 samples, resulting in 8,170 such genes" [1500 less than the number of DAP3-bound genes] (lines 405-406). Thus, the authors are indicating that DAP3 binds a significant majority (perhaps nearly all?) of expressed genes in EC109 cells. A similar vein runs through the differential splicing analysis – the authors identify 7,400 and 11,820 DAP3-regulated splicing events; this strikes me as an extraordinarily high number of significant events (for example, higher than all but one RBP in the recent ENCODE RBP shRNA knockdown dataset [PMID 32728246], with RBPs with similar numbers of altered events all being core spliceosomal machinery factors like AQR, U2AF2, SF1, etc).

As an initial concern, I feel that the manuscript somewhat downplays the above. With the splicing events, for example – 6140 altered SE events in KYSE180 cell lines is not only a large number of significantly differentially spliced SEs, it strikes me as a high number of cassette exons in general to confidently observe from ~50M read RNA-seq. Thus I think it would be helpful to include additional analyses here – are many of these exons either not included (PSI=0) or constitutive (PSI=100) in wild-type cells? It seems to me that either the DAP3 knockdown is causing mis-splicing of a large number of normally constitutive (or absent) exons, or DAP3 knockdown is significantly altering the vast majority of cassette exons in wild-type cells – either answer would be intriguing, but it would aid interpretation to know which is true. In this vein, I also think having some explicit discussion of what is abundantly expressed but not bound by DAP3 would be helpful – are there many such genes? Do they tend to have a specific pattern (are they single-exon genes, lack alternative exons, represent essential machinery factors, etc)?

This also creates challenges for interpreting the strength of later analyses. For example, lines 137-139 "Based on this observation, it is likely the majority of DAP3-regulated alternative splicing events are mediated through binding of DAP3 to the RNA transcripts." – since DAP3 likely binds more than half of expressed genes in this cell type, is this overlap actually significant? Since alternative splicing events are also likely to be detected at more abundant genes (simply due to higher likelihood of sufficient junction-spanning reads for quantitation), if DAP3 binds 2/3 of highly expressed genes, observing peaks at half of DAP3 knockdown-differentially spliced genes would suggest no correlation between DAP3 binding and differential splicing. Having proper background or randomization for these overlaps (as well as explicit definition of how large the 'expressed but unbound' sets actually are) would help to clarify whether these are actually enriched or reflect the expected frequency.

As an alternate path – the authors use a single cutoff for eCLIP peak calling throughout the manuscript, but I think it could be helpful to understand whether there is a correlation between eCLIP enrichment and alternative splicing (in Fig. 2c for example), by isolating targets based on fold-enrichment or other metric to define 'strong' binding sites. I also think it would be helpful to show browser read density tracks for the key events being used in the manuscript – the WSB1 event, for example, forms the basis for all of Fig. 6; it would be helpful to see the raw data and to what degree of DAP3 enrichment is observed at this event.

Similarly, in Fig. 6a –how often do alternative exons in general show this kind of pattern (in other words, are DAP3-regulated events more likely to be dysregulated in tumors, or does the

observation of this many events simply reflect that DAP3 seems to regulate a large number of events)? In Fig. 6d-e, is 14% more or less than expected by chance?

As one other major concern - I think the co-interaction data with NONO and SFPO are intriguing (and well-validated); however, I do have concern with how this feeds back to the DAP3 eCLIP analysis. Are the authors confident that the eCLIP of DAP3 is not co-immunoprecipitating NONO and/or SFPO? I think in this case, it would be prudent to at least show the co-IP western blot experiment performed with the same wash conditions as done in eCLIP to confirm a lack of co-immunoprecipitation in those conditions; however, better would be to show the standard CLIP RNA visualization experiment to confirm that the RNA being isolated during the eCLIP is being bound by DAP3 and not by co-immunoprecipitated NONO or SFPO.

Other comments:

- Why were the examples in Fig. 1c chosen? It strikes me as strange to show 4 example eCLIP tracks in the manuscript, all 4 of which show binding patterns that don't coincide with the 'major' pattern of exonic, CDS enrichment shown in Fig. 1a-b.
- Similarly, why were the examples in Fig. 1d chosen – are these known sites? I would find it helpful to show browser tracks for these examples as well (at least in supplementary) to be able to understand why the experiment in Fig. 1d is confirming high-confidence binding sites
- For the motif analysis in Fig. 1f – how this is done is unclear (the manuscript just cites HOMER) – what is the background set being used? In particular, is the 5'ss-like motif seen in exonic sequences (or is this motif appearing because 5'ss sequences are being included in peak calls that extend past the exon)?
- Do the authors have an idea why WSB1 PSI in control cells seems to vary significantly across experiments but not within an experiment (PSI=20 in EV in Fig. 3g but PSI=5 in EV+scr in Fig. 3h)?
- In Fig. 5f – is this figure only showing events with same-direction change? Is this more than expected by chance (or are there an equal number of events that are discordant between the two KDs)?
- Fig. 6f seems to not really be discussed in the text
- In Fig 7 i-m, I think it is hard to draw strong conclusions without a WT + WSB1 over-expression control – does WSB1 over-expression alone show these same patterns?
- I would recommend toning down the language in some places in the discussion – statements like 'As DAP3 is positioned at the top of splicing cascades composed of hundreds of RBPs' are really not strongly backed up by data (and aren't necessary for the manuscript to stand on its own)

Minor comments:

- The details of the antibody used for eCLIP should be listed in that section (I assume it's the same antibody as listed elsewhere in the Methods, but it's not explicitly stated as the one used for eCLIP)
- 'Regulated' is missing an e in label for Fig. 2h

Reviewer #2 (Remarks to the Author):

This is an interesting manuscript which suggests a role for DAP3 in RNA splicing. While the work shown is generally of high quality, a number of points need to be addressed to solidify and clarify the findings as follows:

- It is difficult to understand the relative contribution of a direct or indirect role of DAP3 in RNA splicing relative to prior publications from this group identifying a role for DAP3 in RNA editing. RNA editing in nascent RNA could impact pre-mRNA splicing. Can the authors reconcile these points?
- Does DAP3 bind RNA directly? Does DAP3 have known RNA binding domains?
- How do the DAP3 RNA binding motifs suggested in Figure 1f compare to the nucleotides enriched or depleted in cassette exons associated with DAP3 suppression in Figure 2a-b? It is possible that DAP3 binding is not linked to splicing regulated by DAP3.
- Related to the above points, the authors do not address if the RNA binding activity of DAP3 is required for splicing. This could be addressed by generating minigenes of some of the candidate

DAP3-associated splicing events shown in the figure and evaluating requirement of DAP3 binding sites in regulating splicing.

-The Abstract needs to be revised. The first sentence is very awkwardly written. In the 2nd sentence, it is not clear what the authors mean by "cis mutations" (I assume the authors mean mutations within genes impacting their own splicing in cis) but this should be clearer. Equally important, the rationale and motivation for investigating a role for DAP3 in RNA splicing is not provided in the Abstract at all.

Point-by-Point Responses

Reviewer #1:

This manuscript by Han et al. describes a deep genomic and molecular exploration of the regulatory networks downstream of DAP3, combining target identification by eCLIP and re-analysis of knockdown/RNA-seq to uncover the role of DAP3 in broadly regulating alternative splicing. I think the results here are quite intriguing and suggest a significant and unappreciated importance of DAP3 in splicing regulation, with links to cancer-relevant altered splicing programs.

1. My main area of concern regards the extremely high number of targets (9,699 genes) seen for DAP3, and what this means for the interpretation of some of the resulting analyses. The authors note in the discussion that this is ~1/3 of total human genes, but that includes many tissue-specific and other poorly expressed genes. Indeed the authors note in the methods for eCLIP analysis that to define expressed genes for use in various analyses, “the expressed genes were defined as having TPM ≥ 1 in each of 10 samples, resulting in 8,170 such genes” [1500 less than the number of DAP3-bound genes] (lines 405-406). Thus, the authors are indicating that DAP3 binds a significant majority (perhaps nearly all?) of expressed genes in EC109 cells. A similar vein runs through the differential splicing analysis – the authors identify 7,400 and 11,820 DAP3-regulated splicing events; this strikes me as an extraordinarily high number of significant events (for example, higher than all but one RBP in the recent ENCODE RBP shRNA knockdown dataset [PMID 32728246], with RBPs with similar numbers of altered events all being core spliceosomal machinery factors like AQR, U2AF2, SF1, etc).

- We note the reviewer’s concern here and apologise for insufficient clarifications in the original manuscript. We would like to clarify that the eCLIP-seq analysis of DAP3-bound genes (**Fig. 1a**) is independent from the metagene analysis which was to understand the distribution of DAP3-binding peaks on transcripts (**Fig. 1b**), and thus, they are not meant to be overlapped/compared directly, due to the following reasons:

1) In **Fig. 1a**, 9,699 DAP3 target genes were identified in two biological replicates of DAP3 eCLIP samples, and they represent any gene that was bound by DAP3. In this analysis, the filtering was applied on the binding strength (fold enrichment of peaks and p-value) but not on the expression level and we assumed that binding requires expression of the target gene. This primary set of 9,699 genes has been used throughout the manuscript for different analyses. Since the idea here is to get a comprehensive list of DAP3 eCLIP peaks to investigate all possible binding events including those occurring in non-coding regions (e.g., introns and intergenic regions), this primary set of genes include not only highly expressed genes but genes which are lowly expressed.

2) In **Fig. 1b**, we intended to understand the distribution of DAP3-binding peaks on transcripts by metagene analysis. To increase the reliability of this analysis, we only considered “expressed genes”. The definition of “expressed genes” here is those genes have TPM (transcripts per million) ≥ 1 in each of 10 RNA-Seq datasets of EC109 cells generated by our team previously. Poorly expressed genes (TPM <1) and genes that are not consistently expressed in EC109 cells (i.e., genes were identified from some EC109 RNA-Seq datasets but not the others) were excluded from the analysis as they could potentially blur the actual binding pattern. Moreover, only the longest isoform was considered as the representative transcript per gene, which means a poorly expressed such transcript was excluded regardless of abundant expression of other shorter transcripts. In addition, peaks that failed to be mapped to these genes would not be considered in the final plot (**Fig. 1b**). Please note that these 8,170 genes were only used for the metagene analysis to understand the overall pattern of DAP3 peak distribution across the coding region defined as “metagene profile” (Van Nostrand et. al., *Genome Biol.*, 2020).

- To answer the reviewer’s question on what percentage of expressed genes were bound/unbound by DAP3, we first defined genes with any of isoforms passing the cut-off of TPM ≥ 1 in each of 10 RNA-Seq datasets of EC109 cells as “expressed genes” (rather than only the longest isoform included for the analysis in **Fig. 1b**) and we identified a total of 11,772 expressed genes in EC109 cells. We found 61%

(7,205/11,772) of expressed genes is bound by DAP3, suggesting DAP3 binds extensively to a large portion of the transcriptome.

expressed gene (TPM \geq 1)	number	percentage
DAP3 bound	7,205	61%
DAP3 unbound	4,567	39%
total	11,772	100%

2. As an initial concern, I feel that the manuscript somewhat downplays the above. With the splicing events, for example – 6140 altered SE events in KYSE180 cell lines is not only a large number of significantly differentially spliced SEs, it strikes me as a high number of cassette exons in general to confidently observe from ~50M read RNA-seq. Thus I think it would be helpful to include additional analyses here – are many of these exons either not included (PSI=0) or constitutive (PSI=100) in wild-type cells? It seems to me that either the DAP3 knockdown is causing mis-splicing of a large number of normally constitutive (or absent) exons, or DAP3 knockdown is significantly altering the vast majority of cassette exons in wild-type cells – either answer would be intriguing, but it would aid interpretation to know which is true. In this vein, I also think having some explicit discussion of what is abundantly expressed but not bound by DAP3 would be helpful – are there many such genes? Do they tend to have a specific pattern (are they single-exon genes, lack alternative exons, represent essential machinery factors, etc)?

- We thank the reviewer for raising these questions. We would like to clarify approximately **200M** (not ~50M as the reviewer mentioned) clean reads (read length 100bp) per sample were obtained from our strand-specific RNA-Seq (Please see the table below). Such a high sequencing depth enabled us to identify much more high-confidence splicing events using rMATS.

Sample Name	Read length(bp)	Clean Reads	Clean bases
EC109 scr	100	217,960,698	21,796,069,800
EC109 DAP3 sh1	100	186,106,402	18,610,640,200
EC109 DAP3 sh2	100	190,710,500	19,071,050,000
K180 scr	100	190,352,846	19,035,284,600
K180 DAP3 sh1	100	212,905,244	21,290,524,400
K180 DAP3 sh2	100	210,597,642	21,059,764,200

To address the reviewer’s question on whether normally constitutive/absent exons or alternative exons are more affected by DAP3, we first analysed the rMATS data of EC109 and KYSE180 Scr control cells and stratified these SE events (108,399 and 110,895 in EC109 and KYSE180, respectively; read coverage \geq 20) into groups based on their PSI values (**Rebuttal Fig. 1a**). Approximately 5% of these SE events (EC109: 5,714; KYSE180: 5,330) have PSI=0% (normally absent), while approximately 20% of these SE events (EC109: 24,737; KYSE180: 28,636) have PSI=100% (normally constitutive). We then analysed the proportion of SE events which were regulated by DAP3 in each group and found that < 1% of SE events in “PSI=0%” group and < 3% of SE events in “PSI=100%” were regulated by DAP3 (**Rebuttal Fig. 1b**). These results suggested that only a very small portion of normally constitutive/absent exons are regulated by DAP3. On the other hand, approximately 15%~25% of SE events detected in each of the remaining groups were regulated by DAP3. These findings indicated that compared to normally constitutive/absent SE events, those with basal PSI values ranging from 20% to 80% are preferentially affected by DAP3.

In addition, we have done a comparative analysis of gene structure between genes expressed and bound by DAP3 and those expressed but not bound by DAP3 (**Rebuttal Fig. 1c**). We did not observe any significant difference in the number of exons or isoforms between DAP3-bound and unbound genes.

These data have been included as the **Supplementary Fig. 2a, b** and **Supplementary Fig. 4a, b**.

Rebuttal Fig. 1. (a) Bar charts showing the number of SE events in the indicated groups based on the basal PSI values in EC109 and KYSE180 scramble control cells. (b) Bar charts showing the percentage of DAP3-regulated SE events in the indicated groups. (c, d) Bar charts showing the frequency of DAP3-bound or unbound expressed genes (y-axis) with the indicated number of exons (c) or isoforms (d) per gene (exon counts or isoform counts; x-axis) in EC109 cells.

3. This also creates challenges for interpreting the strength of later analyses. For example, lines 137-139 “Based on this observation, it is likely the majority of DAP3-regulated alternative splicing events are mediated through binding of DAP3 to the RNA transcripts.” – since DAP3 likely binds more than half of expressed genes in this cell type, is this overlap actually significant? Since alternative splicing events are also likely to be detected at more abundant genes (simply due to higher likelihood of sufficient junction-spanning reads for quantitation), if DAP3 binds 2/3 of highly expressed genes, observing peaks at half of DAP3 knockdown-differentially spliced genes would suggest no correlation between DAP3 binding and differential splicing. Having proper background or randomization for these overlaps (as well as explicit definition of how large the ‘expressed but unbound’ sets actually are) would help to clarify whether these are actually enriched or reflect the expected frequency.

- We note the reviewer’s concern here. To assess the significance of DAP3 binding on differential splicing regulation, we calculated the ratio of DAP3 eCLIP peaks mapped to transcripts with DAP3-regulated or non-regulated alternative splicing (AS). We first identified DAP3-regulated and non-regulated genes as described below:

1. “DAP3-regulated genes” group (n=3,262): Genes with differentially spliced AS events caused by DAP3 knockdown; and
2. “Non-regulated genes” group (n=9,828): All other genes undergoing alternative splicing identified by rMATs in the same sample but not regulated by DAP3 as they did not pass the thresholds for our differential analysis.

We next calculated the proportions of DAP3-bound genes in these two groups for comparison. We had reported in the manuscript that 67% (2,180/ 3,262) of DAP3-regulated genes were also bound by DAP3, and here we found that 52% (5,121 / 9,828) of non-regulated genes were also bound by DAP3. This shows that DAP3 indeed binds to its target genes for splicing regulation by 15% more than the rest of the genes undergoing AS but not regulated by DAP3 (i.e., background, $p < 2.2e-16$ by Fisher’s test) (revised **Fig. 2c**), suggesting gene undergoing DAP3-regulated splicing is significantly enriched with DAP3 binding peaks.

To account for the two factors “expression levels” and “sample size” in the comparison, we did a further randomization analysis. First, we checked the expression distributions of the two groups and found there was no significant difference in expression levels of DAP3-regulated and non-regulated AS genes (Please see the table below and **Rebuttal Fig. 2a**). Next, we randomly selected 3,262 genes out of 9,828 non-regulated AS genes, followed by the analysis of the number of bound genes (the analysis was repeated 100 times). When compared to 2,180 bound genes in “DAP3-regulated AS genes” group, an average of ~1,700 bound genes found in “Non-regulated AS genes” group confirmed the enrichment of DAP3 binding to its splicing targets (**Rebuttal Fig. 2b**). All the relevant information and data have been included in the revised **Supplementary information** and **Supplementary Fig. 4c and d**.

Group	Expression Distribution						
	n	Min	1st Quarter	Median	Mean	3rd Quarter	Max
DAP3-regulated genes	3,262	0	7.101	16.364	63.994	40.268	7115.774
Non-regulated genes	9,828	0	3.169	14.711	63.967	44.086	7062.773

Rebuttal Fig. 2. (a) Expression levels (TPM) of genes in the “DAP3-regulated genes” and “Non-regulated genes” groups. Data are presented as box plots with median (horizontal line), mean (‘+’), 25–75 percentile (box), and min to max values (whisker) for each group (Unpaired Welch’s *t*-test; ns, not significant). (b) Bar chart showing the frequency distribution of gene sets (n=100) with the indicated number of DAP3 eCLIP peaks. Each gene set comprised randomly selected 3,262 genes from the 9,828 non-regulated genes.

4. As an alternate path – the authors use a single cutoff for eCLIP peak calling throughout the manuscript, but I think it could be helpful to understand whether there is a correlation between eCLIP enrichment and alternative splicing (in Fig. 2c for example), by isolating targets based on fold-enrichment or other metric to define ‘strong’ binding sites. I also think it would be helpful to show browser read density tracks for the key events being used in the manuscript – the WSB1 event, for example, forms the basis for all of Fig. 6; it would be helpful to see the raw data and to what degree of DAP3 enrichment is observed at this event.

- Thank the reviewer for giving valuable suggestions. To address this point, we divided DAP3-bound genes in each eCLIP biological replicate based on the fold enrichment of their eCLIP peaks into 4 groups (≥ 4 , ≥ 8 , ≥ 12 and ≥ 16 fold) and checked the percentage of bound genes which underwent or did not undergo DAP3-regulated AS among all DAP3-bound genes. We found that the binding affinity of DAP3 (represented by eCLIP fold enrichment) does not correlate with the strength of splicing regulation as the proportion of bound genes with DAP3-regulated AS remains within the range of 20% to 25% among all DAP3-bound genes (**Rebuttal Fig. 3**). This indicates an increased binding affinity of DAP3 does not potentiate its splicing regulation. These data have been included in the revised **Supplementary Fig. 4e**.

In addition, as the reviewer suggested, we added the eCLIP browser tracks for key events in the revised **Supplementary Fig. 7 (Rebuttal Fig. 4)**. The detailed information about the position, enrichment fold and *p*-value of each eCLIP peak was included in the revised **Supplementary Table 1**.

Rebuttal Fig. 4. DAP3 binding peaks on *WSB1*, *SSBP3* and *RHBDD2* gene, from the eCLIP-seq data. (a-c) IGV browser tracks of the DAP3 eCLIP peaks spanning exon5 to exon6 of *WSB1* gene (a), exon5 to exon7 of *SSBP3* gene (b), and exon1 to exon3 of *RHBDD2* gene (c).

5. Similarly, in Fig. 6a –how often do alternative exons in general show this kind of pattern (in other words, are DAP3-regulated events more likely to be dysregulated in tumors, or does the observation of this many events simply reflect that DAP3 seems to regulate a large number of events)? In Fig. 6d-e, is 14% more or less than expected by chance?

- Thank the reviewer for raising these questions. We would like to clarify that by analysing TCGA datasets, we did not intend to show DAP3-regulated events are more likely to be dysregulated in tumors compared to other alternative splicing events. The purpose of showing the data in **Fig. 6a** was to highlight 1) the DAP3-regulated AS events could be detected in patients' tumors (not just in cancer cell lines), and 2) these events are dysregulated in multiple tumor types. Aberrant splicing is associated with numerous cancers, and it is generally caused by mutations in *cis*- and *trans*-splicing regulatory elements and altered expression of splicing factors and regulators. Besides DAP3, there are many other splicing regulatory proteins which contribute to dysregulated splicing in tumors. Therefore, with our current data, we are not able to conclude DAP3-regulated events are more likely to be dysregulated in tumors. However, we would like to keep it for our further investigation.

It is not surprising that we did not see a high overlap of DAP3-regulated AS events between our EC109 and the TCGA ESCA RNA-Seq datasets, with the following reasons:

- 1) There are many AS events present in tumor samples which may be regulated by other splicing factors or regulators.
- 2) Different from cancer cell lines, tumor tissues are known to be heterogeneous and consist of different cell types including tumor cells (or sub-clones) and various microenvironmental cell types such as infiltrating immune cells and stromal cells.
- 3) DAP3 TCGA RNA-Seq datasets have much lower sequencing depth compared to our EC109 datasets and we only included AS events which were found to be differentially spliced in all 13 tumors for the analysis, which explain why only 3,811 AS events were detected in ESCA tumors (5-fold less than the number of DAP3-regulated AS events from our EC109 RNA-Seq data).

6. As one other major concern - I think the co-interaction data with NONO and SFPQ are intriguing (and well-validated); however, I do have concern with how this feeds back to the DAP3 eCLIP analysis. Are the authors confident that the eCLIP of DAP3 is not co-immunoprecipitating NONO and/or SFPQ? I think in this case, it would be prudent to at least show the co-IP western blot experiment performed with the same wash conditions as done in eCLIP to confirm a lack of co-immunoprecipitation in those conditions; however, better would be to show the standard CLIP RNA visualization experiment to confirm that the RNA being isolated during the eCLIP is being bound by DAP3 and not by co-immunoprecipitated NONO or SFPQ.

- We note the reviewer's concern. To address this, we performed co-IP experiment with the same wash conditions as used in the eCLIP assay and confirmed neither SFPQ nor NONO was detected in DAP3 immunoprecipitates (**Rebuttal Fig. 5**). As the editor suggested, we also checked two well-known spliceosomal proteins U2AF35 and U2AF65 in the DAP3 immunoprecipitates and found no presence of these two proteins in the IP eluates (**Rebuttal Fig. 5**). All these data suggested that when eCLIP-Seq was carried out, the wash condition was harsh enough to remove DAP3 interactors and other RBPs which may potentially bind to the target RNAs. We have included these data and the relevant information in the revised **Supplementary Information** and **Supplementary Fig. 1b**.

Rebuttal Fig. 5. Western blot analyses of SFPQ, NONO, U2AF35 and U2AF65 proteins in the DAP3 eCLIP samples. DAP3-1 and DAP3-2 are two eCLIP replicates.

Other comments:

1. Why were the examples in Fig. 1c chosen? It strikes me as strange to show 4 example eCLIP tracks in the manuscript, all 4 of which show binding patterns that don't coincide with the 'major' pattern of exonic, CDS enrichment shown in Fig. 1a-b.

- Our eCLIP analysis demonstrated DAP3 binds to both non-coding RNAs (~6%) and protein-coding RNAs (~80%), we therefore selected two representative non-coding RNAs (*NEAT1* and *MALAT1*) and two protein-coding RNAs (*ARHGEF16* and *TARDBP*) which have multiple DAP3-binding peaks. Another reason why we chose *ARHGEF16* and *TARDBP* was that although DAP3 mainly binds to exonic regions, it also binds to intronic regions, 5'UTRs and 3'UTRs and we could see multiple peaks at intronic region and 3' UTR of *ARHGEF16* and *TARDBP* transcripts.

2. Similarly, why were the examples in Fig. 1d chosen – are these known sites? I would find it helpful to show browser tracks for these examples as well (at least in supplementary) to be able to understand why the experiment in Fig. 1d is confirming high-confidence binding sites.

- We thank the reviewer for pointing this out. The *SCYL1* probe (chr11:65293060-65293102; GUCCAUCUUCGUCUAUGAUGUGAAGCCUGGCGCGGAAGAGC) contains CAUCUUC which matches with the second ranked DAP3 RNA-binding motifs shown in **Fig. 1d**. The *ATAD3A* probe (chr1:1469369-1469421; AUGCUGUCCAGCAGCACCAGCAGAAGAUGUGCUGGCUGAAGGCGGAAGGGCC) contains GAAGAU – the top-ranking DAP3 RNA-binding motif (**Fig. 1d**). The browser tracks for these two targets and the location of the probe sequence (indicated by blue bar) are shown in the **Rebuttal Fig. 6**. We have added these data into the revised **Supplementary Fig. 3b**.

Rebuttal Fig. 6. IGV browser tracks of the DAP3 eCLIP peaks spanning the genomic loci of *SCYL1* and *ATAD3A* gene. The positions of RNA probe sequences used in REMSA are indicated.

3. For the motif analysis in Fig. 1f – how this is done is unclear (the manuscript just cites HOMER) – what is the background set being used? In particular, is the 5'ss-like motif seen in exonic sequences (or is this motif appearing because 5'ss sequences are being included in peak calls that extend past the exon)?

- Thank the reviewer for pointing it out. We used “findMotifsGenome.pl” function in HOMER for “de novo” motif enrichment with the flag “-rna”. The background set consists of randomly selected sequences from the reference genome, corrected for sequence content (GC content and other binucleotide composition, see “<http://homer.ucsd.edu/homer/motif>” and “<http://homer.ucsd.edu/homer/motif/rnaMotifs.html>” for details). We have added this information into the revised “Methods”. We would like to apologise that we inadvertently mixed up the motifs shown in the **previous Fig. 1f** (motif analysis using all DAP3 eCLIP peak sequences) and **Supplementary Fig. 4g** (motif analysis using the sequences of DAP3 eCLIP peaks within the region from the upstream to

downstream constitutive exon of an alternatively spliced event (shown in **Supplementary Fig. 4h**). We have corrected the mistake in this submission.

Theoretically, the motifs can be found anywhere within the central 200 base pairs of input sequences (peak regions). We did not apply any restriction on the boundaries of motif search. The 5'ss-like motif was enriched because many DAP3 eCLIP peaks were mapped to the boundary of the 5'ss as illustrated in the **rebuttal Fig. 7** below. We have also added these data into the revised **Supplementary Fig. 3a**.

Rebuttal Fig. 7. IGV browser tracks of the DAP3 eCLIP peaks spanning the 5' splice site junction with AGGUAAGU motifs on *ZC3H12A*, *PRKAB2*, *RILPL1* and *EED* gene. DAP3-binding motif AGGUAAGU is indicated by box.

4. Do the authors have an idea why *WSB1* PSI in control cells seems to vary significantly across experiments but not within an experiment (PSI=20 in EV in Fig. 3g but PSI=5 in EV+scr in Fig. 3h)?

- We thank the reviewer for pointing this out. We would like to clarify that cells used for the overexpression of empty vector control (EV), SFPQ, or NONO in **Fig. 3g** and **Fig. 3h** were different. In **Fig. 3g**, parental (untreated) cells were used for overexpression; while cells mentioned in **Fig. 3h** underwent puromycin selection to establish stable Scr cells prior to the overexpression experiment, which may explain why the basal *WSB1* PSI value varies across different experiments. However, regardless of the changes in the basal PSI, upon the same treatment in an experiment, we could still observe the same trend of PSI changes.

5. In Fig. 5f – is this figure only showing events with same-direction change? Is this more than expected by chance (or are there an equal number of events that are discordant between the two KDs)?

- Yes, events with same-direction change were included in this heatmap because we aimed to detect DAP3-regulated splicing events due to DAP3-mediated RBM6 downregulation. To exclude the probability that these co-regulated events were observed by chance, we perform rescue experiments in **Fig. 5h** to confirm that restoring RBM6 expression in DAP3 depleted cells could rescue the splicing changes. We also observed 99 events that are discordant between the two KDs. Because splicing regulation often involves numerous RBPs and we showed DAP3 regulated splicing of hundreds of RBPs, it is possible that the regulation of these discordant splicing events involves other DAP3-regulated RBPs other than RBM6 alone. Therefore, the overall effects of these discordant splicing events after DAP3 depletion were not dominated by RBM6 downregulation. This part of data has been discussed in the resubmitted manuscript.

6. *Fig. 6f seems to not really be discussed in the text.*

- We thank the reviewer for pointing this out. We have added relevant discussion in revised manuscript.

7. *In Fig 7 i-m, I think it is hard to draw strong conclusions without a WT + WSB1 over-expression control.*

– *does WSB1 over-expression alone show these same patterns?*

- As suggested by the reviewer, we have included an additional WT+WSB1 control and repeated the *in vitro* culture-based and *in vivo* tumorigenicity assays in the revised **Fig. 7h-k**. WSB1 overexpression in WT cells increased tumorigenicity, indicating its oncogenic function. Since we have demonstrated *DAP3* KO caused AS-NMD of *WSB1* to suppress WSB1 expression, we further tested whether restoring WSB1 expression in *DAP3* KO cells could rescue the reduced tumorigenicity caused by *DAP3* KO. As demonstrated in **Fig. 7h-k**, restoring WSB1 expression in *DAP3* KO cells significantly rescued the reduced tumorigenicity caused by *DAP3* KO, suggesting that *DAP3* KO represses tumorigenesis at least partially through promoting AS-NMD of *WSB1* gene.

8. *I would recommend toning down the language in some places in the discussion – statements like ‘As DAP3 is positioned at the top of splicing cascades composed of hundreds of RBPs’ are really not strongly backed up by data (and aren’t necessary for the manuscript to stand on its own)*

- We agree with the reviewer’s suggestion and we have amended the sentence in the discussion.

Minor comments:

1. *The details of the antibody used for eCLIP should be listed in that section (I assume it’s the same antibody as listed elsewhere in the Methods, but it’s not explicitly stated as the one used for eCLIP)*

- Yes, it is the same antibody listed in method part.

2. *‘Regulated’ is missing an e in label for Fig. 2h*

- Thanks for pointing out and we have corrected it.

Reviewer #2:

This is an interesting manuscript which suggests a role for DAP3 in RNA splicing. While the work shown is generally of high quality, a number of points need to be addressed to solidify and clarify the findings as follows:

1. It is difficult to understand the relative contribution of a direct or indirect role of DAP3 in RNA splicing relative to prior publications from this group identifying a role for DAP3 in RNA editing. RNA editing in nascent RNA could impact pre-mRNA splicing. Can the authors reconcile these points?

- Thank the reviewer for making this valuable suggestion. To address this point, we intersected a total of 1,109 and 2,322 DAP3-regulated editing sites (from our previously published paper) with 7,400 and 11,820 DAP3-regulated splicing events obtained from EC109 and KYSE180 RNA-Seq samples, respectively. We found only 1~2% of AS events having editing sites detected in the region from the upstream to downstream constitutive exon of alternatively spliced event as shown in **Rebuttal Fig. 8**. Although this does not exclude the possibility that DAP3-regulated RNA editing in nascent RNAs could impact pre-mRNA splicing, this does not stand as a dominant mechanism in our case. We have also added these data into the revised **Supplementary Fig. 4h and i**.

Rebuttal Fig. 8. (a) Schematic diagram illustrating 5 types of AS events. Editing sites residing in the indicated region (red arrow; its boundaries are indicated red vertical lines) were included for the analysis described in (b). (b) Pie charts showing the proportion of DAP3-regulated AS events with or without DAP3-regulated A-to-I editing sites within the indicated region.

2. Does DAP3 bind RNA directly? Does DAP3 have known RNA binding domains?

- Our eCLIP-seq and REMSA analyses confirmed DAP3 binds RNA directly *in vivo* and *in vitro* (**Fig. 1a, c and e**). The harsh wash conditions used in the eCLIP experiments ensures the detection of direct RNA binding sites of RNA-binding proteins. To further confirm this, we performed co-IP experiment with the same wash conditions as used in the eCLIP assay and confirmed two DAP3-interacting proteins SFPQ and NONO were not detected in DAP3 immunoprecipitates (the revised **Supplementary Fig. 1b**). As the editor suggested, we also checked two well-known spliceosomal proteins U2AF35 and U2AF65 in the DAP3 immunoprecipitates and found no presence of these two proteins in the IP eluates (the revised **Supplementary Fig. 1b**). All these data suggested that when eCLIP-Seq was carried out, the wash condition was harsh enough to remove potential DAP3 interactors, confirming the reliability of our DAP3 eCLIP-Seq data and more importantly the RNA binding capability of DAP3. So far, no known RNA binding domain of DAP3 has been reported.

3. How do the DAP3 RNA binding motifs suggested in Figure 1f compare to the nucleotides enriched or depleted in cassette exons associated with DAP3 suppression in Figure 2a-b? It is possible that DAP3 binding is not linked to splicing regulated by DAP3.

4. Related to the above points, the authors do not address if the RNA binding activity of DAP3 is required for splicing. This could be addressed by generating minigenes of some of the candidate DAP3-associated splicing events shown in the figure and evaluating requirement of DAP3 binding sites in regulating splicing.

- Since these 2 comments are related, we would like to address them together. To address the reviewer's comment #3, we performed motif enrichment analysis on cassette exons from DAP3-regulated SE events in EC109 cells. We used the same tool and parameters (HOMER, findMotifsGenome.pl, -rna flag) for the analysis. The top 3 motifs are given below and was also included as the revised **Supplementary Fig. 4f**. We observed the top motif "AGGUAAGU" analysed from the cassette exons is highly similar to the third ranked DAP3 RNA-binding motif analysed from eCLIP peaks; and the third ranked motif from cassette exons partially matches with the top-ranking DAP3 RNA-binding motif (**Rebuttal Fig. 9**).

To address the reviewer's question on whether DAP3 binding is not linked to DAP3-regulated splicing, we went on to assess the significance of DAP3 binding on differential splicing regulation, we calculated the ratio of DAP3 eCLIP peaks mapped to transcripts with DAP3-regulated or non-regulated AS. We first identified DAP3-regulated and non-regulated genes as described below:

1. "DAP3-regulated genes" group (n=3,262): Genes with differentially spliced AS events caused by DAP3 knockdown; and
2. "Non-regulated genes" group (n=9,828): All other genes undergoing alternative splicing identified by rMATs in the same sample but not regulated by DAP3 as they did not pass the thresholds for our differential analysis.

We next calculated the proportions of DAP3-bound genes in these two groups for comparison. We had reported in the manuscript that 67% (2,180/ 3,262) of DAP3-regulated genes were also bound by DAP3, and here we found that 52% (5,121 / 9,828) of non-regulated genes were also bound by DAP3. This shows that DAP3 indeed binds to its target genes for splicing regulation by 15% more than the rest of the genes undergoing AS but not regulated by DAP3 (i.e., background, $p < 2.2e-16$ by Fisher's test) (revised **Fig. 2c**), suggesting regulation of splicing by DAP3 is more likely to be mediated through binding.

As requested by the reviewer in the comment #4, more experimental evidence should be provided to support the conclusion that DAP3 binding to target RNA is required for its splicing regulation. To this end, we first constructed a wildtype (wt) minigene consisting of *WSB1* exons 5-6 and intervening introns. Based on the identified DAP3 eCLIP peaks on *WSB1* gene, we next generated two deletion mutant minigenes by deleting a 58bp DAP3 binding sequence on *WSB1* exon E6a/b (del mut1) or a 9bp GATGAAGTA DAP3 binding motif (del mut2) (**Rebuttal Fig. 10a**). First, we found depletion of DAP3 led to the inclusion of previously unannotated exon E6a and E6b in the wt minigene (**Rebuttal Fig. 10b**), consistent with the splicing change of endogenous *WSB1* upon DAP3 depletion (**Fig. 2f**). However, such splicing changes were not observed in *WSB1* transcripts derived from del mut1 or del mut2 minigene upon DAP3 knockdown (**Rebuttal Fig. 10b**). Next, RNA pulldown assays using RNA probe consisting of the *WSB1* wt exon E6b (E6b wt), the 58nt DAP3 binding sequence-depleted exon E6b (E6b del mut1) or the 9nt GATGAAGTA binding motif- depleted exon E6b (E6b del mut2) further confirmed the deleted sequences were required for DAP3 binding (**Rebuttal Fig. 10c**). This indicates that as least for a subset of genes undergoing DAP3-regulated splicing such as *WSB1*, the binding of DAP3 to its target RNA is required for its splicing regulation. We have included these new data into **Fig. 2h-j**.

In addition, we would like to clarify that we detected DAP3 binding sites in 67% of genes undergoing DAP3-regulated AS, meaning there are 33% of genes undergoing DAP3-regulated AS but having no DAP3 binding (**Fig. 2c**). In fact, we showed that DAP3 can indirectly regulate splicing via altering splicing and expression of splicing factors (**Fig. 5**).

Rebuttal Fig. 9. (a) Top 3 enriched motifs in sequences of DAP3-regulated cassette exons. (b) Alignment of DAP3 eCLIP motifs and DAP3-regulated cassette exons motifs.

Rebuttal Fig. 10. (a) IGV browser tracks of the DAP3 eCLIP peaks spanning exons 5-6 and intervening introns of *WSB1* gene. Schematic diagram illustrates the genomic fragments inserted into the wildtype (wt) and mutant *WSB1* splicing minigenes. del mut1, lacking a 58bp DAP3 binding sequence in exon E6a/b; del mut 2, lacking a 9bp DAP3 binding motif in exon E6a/b. (b) Semi-quantitative RT-PCR analyses of splicing changes of exogenous *WSB1* transcripts derived from the indicated minigenes in DAP3-knockdown and scramble control cells. *ACTB* was used as a housekeep gene internal control. Data are represented as mean \pm s.d. of technical triplicates, statistical significance is determined by unpaired, two-tailed Student's *t*-test (**, $p < 0.01$). (c) RNA pulldown assay detecting the binding of DAP3 to the indicated RNA probe. WB analysis of DAP3 proteins in RNA pulldown (eluate) products and flow-through (FT) fractions.

5. The Abstract needs to be revised. The first sentence is very awkwardly written. In the 2nd sentence, it is not clear what the authors mean by "cis mutations" (I assume the authors mean mutations within genes impacting their own splicing in cis) but this should be clearer. Equally important, the rationale and motivation for investigating a role for DAP3 in RNA splicing is not provided in the Abstract at all.

- Thank the reviewer for pointing this out. We have rewritten the abstract and covered the points mentioned by the reviewer.

Reference

Van Nostrand, Eric L., et al. "Principles of RNA processing from analysis of enhanced CLIP maps for 150 RNA binding proteins." *Genome biology* 21.1 (2020): 1-26.

REVIEWER COMMENTS

Reviewer #1 (Remarks to the Author):

I appreciate the author's careful consideration and detailed responses, and they have addressed most of my concerns

However, I'm a bit confused by the added analysis in Sup. Fig. 4c-e – since the paper focuses on alternative splicing, it seems strange to me to analyze the overlap of DAP3 binding and DAP-3 regulated AS at the gene- rather than event- level (lines 157-172). It seems to me that this should be done asking whether DAP3 is binding either to the altered exon itself or within the upstream – SE – downstream exon region, as I'm not really sure what the usefulness is of knowing that DAP3 binds somewhere within a gene that also contains an exon with altered splicing upon DAP3 depletion.

My other broader writing concern is that the manuscript still (to me) over-uses 'regulate' when the connection between DAP3 binding and functional regulation appears to not be strong enough to make very strong and broad statements. E.g.:

We had reported in the manuscript that 67% (2,180/ 3,262) of DAP3-regulated genes were also bound by DAP3, and here we found that 52% (5,121 / 9,828) of non-regulated genes were also bound by DAP3. This shows that DAP3 indeed binds to its target genes for splicing regulation by 15% more than the rest of the genes undergoing AS but not regulated by DAP3 (i.e., background, $p < 2.2e-16$ by Fisher's test) (revised Fig. 2c), suggesting gene undergoing DAP3-regulated splicing is significantly enriched with DAP3 binding peaks.

I agree that this analysis does show that there is a significant overlap (with the above caveat about gene- vs event- analysis). However, a flipped way to state this would be that there is only a ~1.85 odds ratio for binding to correspond to altered splicing (or, in other words, since $1082/5789 = \sim 19\%$ of unbound genes versus $2180/7301 = \sim 30\%$ of bound genes show differential splicing, that suggests that ~63% of the 'bound and regulated' genes could be false positives).

As such I would recommend changing a lot of the 'DAP3-regulated' terminology to more explicit terms (e.g. 'DAP3 knockout-responsive' or 'DAP3 may modulate splicing'), especially in cases where the claim of regulating is especially tenuous (e.g. lines 198-200 "Moreover, given that one-third (1,082/3,262) of DAP3-regulated alternative spliced genes was not bound by DAP3, it is possible that DAP3 may also regulate splicing through RNA binding-independent mechanisms.")

Other notes:

In looking at the GEO links for the datasets, it appears that the eCLIP data was initially utilized (but not described methodologically) in another publication from this group (PMID 32596459). I think the text and methods section here needs to be clearer with respect to this; I personally don't think it's a problem (the data was not really analyzed or described in that paper), but while the first couple sentences in the results section reference the other publication and obliquely refer to binding analyses in that paper, I think it needs to be more explicitly clear at least in the methods section that this paper is being considered the 'initial' publication and description of that data even though it's already publicly released and utilized in that other paper.

Genome browser figures (1c, 2h, Sup 7) should include the positions of identified peaks

Lines 101-103: No presence of others RBPs in the DAP3 eCLIP elutes confirmed the specificity of the DAP3 eCLIP-Seq analysis (Supplementary information; Supplementary Fig. 1a and b). This is far too strongly stated. As mentioned in the initial review, the standard method in the field to experimentally confirm this degree of specificity of the RBP under study would be to perform RNA visualization (with radiolabeled/fluorescent/biotinylated RNA) in low versus high RNase to show that the recovered RNA is specific to the DAP3 protein size. The supplemental figure referenced is sufficient to validate that the eCLIP is not pulling down the 4 tested proteins, and the overlap with (and overall number of splicing changes seen in) the RNA-seq strongly suggests that

the targets seen are real DAP3 interactions, but the strength of this claim needs to be toned down.

The presence of AGGUAAGU motif (Supplementary Fig. 3a), the 5' splice site consensus sequence in vertebrates, indicates the binding preference of DAP3 for the exon-intron splice junctions. This is probably also too strong – the result indicates that DAP3 has peaks that overlap 5' splice sites, but 'binding preference' would require some sort of binding assay (and as the authors note in Rebuttal Fig 7, it seems more likely that it's primarily binding to the exon and simply having reads that extend across the 5'ss region).

As noted in the first review – the methods section of the manuscript states for eCLIP "Immunoprecipitated using a DAP3 antibody or control IgG". I appreciate that the authors confirmed in their response that the same antibody listed in other sections was used, but I think the manuscript text itself should contain these details (antibody catalog information, concentration used, secondary beads used, etc.) in that section of the text itself.

Minor comments:

Fig 2j is not referenced in the text

Reviewer #2 (Remarks to the Author):

The authors have addressed my prior comments and questions.

Point by point response

Reviewer #1 (Remarks to the Author):

1. I appreciate the author's careful consideration and detailed responses, and they have addressed most of my concerns. However, I'm a bit confused by the added analysis in Sup. Fig. 4c-e – since the paper focuses on alternative splicing, it seems strange to me to analyze the overlap of DAP3 binding and DAP3 regulated AS at the gene- rather than event- level (lines 157-172). It seems to me that this should be done asking whether DAP3 is binding either to the altered exon itself or within the upstream – SE – downstream exon region, as I'm not really sure what the usefulness is of knowing that DAP3 binds somewhere within a gene that also contains an exon with altered splicing upon DAP3 depletion.

*-We thank the reviewer for raising these questions. The new data shown in **Sup. Fig. 4c-e** were used to address the reviewer's concern raised during the last review process on whether genes with detected DAP3 binding (from eCLIP-seq analysis) and DAP3-modulated splicing changes (from RNA-seq analysis) tend to have higher expression level when compared to the rest of genes. As seen in **Sup. Fig. 4c-e**, we showed that 1) there was no significant difference in expression between genes undergoing DAP3-modulated splicing and those without DAP3-modulated splicing; and 2) gene undergoing DAP3-modulated splicing is significantly enriched with DAP3 binding peaks.*

*Next, we would like to respond to the reviewer's query on “*what the usefulness is of knowing that DAP3 binds somewhere within a gene that also contains an exon with altered splicing upon DAP3 depletion*”. Accumulating evidence from previous studies suggested that long-range interactions within the same RNA transcript can be mediated by RBPs and such interactions are heavily involved in RNA processing including RNA splicing (Graveley, 2005; Lovci et al., 2013; Pervouchine et al., 2012; Raker et al., 2009; Sharma et al., 2016). Therefore, in our study, we first analyzed the binding sites of DAP3 on the full-length transcripts rather than DAP3-modulated splicing regions, in order to provide not only an unbiased transcriptome-wide identification of DAP3 binding sites, but also a comprehensive understanding of the relationship between binding and splicing. Secondly, as shown in **Fig. 2d**, the eCLIP peak coverage for all 5 types of DAP3-modulated splicing events indicated a strong tendency of binding to exons rather than introns within the regulated splicing regions. Overall, our eCLIP-seq data, together with RNA-seq data, are powerful data resource to understand the regulatory mechanisms of DAP3-modulated splicing events.*

2. My other broader writing concern is that the manuscript still (to me) over-uses 'regulate' when the connection between DAP3 binding and functional regulation appears to not be strong enough to make very strong and broad statements.

E.g.: We had reported in the manuscript that 67% (2,180/ 3,262) of DAP3-regulated genes were also bound by DAP3, and here we found that 52% (5,121 / 9,828) of non-regulated genes were also bound by DAP3. This shows that DAP3 indeed binds to its target genes for splicing regulation by 15% more than the rest of the genes undergoing AS but not regulated by DAP3 (i.e., background, $p < 2.2e-16$ by Fisher's test) (revised Fig. 2c), suggesting gene undergoing DAP3-regulated splicing is significantly enriched with DAP3 binding peaks.

I agree that this analysis does show that there is a significant overlap (with the above caveat about gene- vs event- analysis). However, a flipped way to state this would be that there is only a ~1.85 odds ratio for binding to correspond to altered splicing (or, in other words, since $1082/5789 = \sim 19\%$ of unbound genes versus $2180/7301 = \sim 30\%$ of bound genes show differential splicing, that suggests that ~63% of the 'bound and regulated' genes could be false positives).

As such I would recommend changing a lot of the ‘DAP3-regulated’ terminology to more explicit terms (e.g. ‘DAP3 knockout-responsive’ or ‘DAP3 may modulate splicing’), especially in cases where the claim of regulating is especially tenuous (e.g. lines 198-200 “Moreover, given that one-third (1,082/3,262) of DAP3-regulated alternative spliced genes was not bound by DAP3, it is possible that DAP3 may also regulate splicing through RNA binding-independent mechanisms.”)

-We greatly appreciate the reviewer’s valuable suggestions which significantly improved the scientific rigor of our study. As suggested, we have changed “DAP3-regulated” to “DAP3-modulated” wherever appropriate throughout the manuscript.

Other notes:

1. In looking at the GEO links for the datasets, it appears that the eCLIP data was initially utilized (but not described methodologically) in another publication from this group (PMID 32596459). I think the text and methods section here needs to be clearer with respect to this; I personally don’t think it’s a problem (the data was not really analyzed or described in that paper), but while the first couple sentences in the results section reference the other publication and obliquely refer to binding analyses in that paper, I think it needs to be more explicitly clear at least in the methods section that this paper is being considered the ‘initial’ publication and description of that data even though it’s already publicly released and utilized in that other paper.

-We thank the reviewer for pointing this out. We have added in-text citation of our previous publication where the eCLIP data was first utilized and also added the GEO accession number at the beginning of the “Results” section. In addition, we added a statement in the “Methods” section, as follows:

“The eCLIP-seq data was initially utilized in our previous study¹¹ (GEO accession number: GSE144318); however, the comprehensive analysis of eCLIP-seq data was provided only in the present study.”

2. Genome browser figures (1c, 2h, Sup 7) should include the positions of identified peaks.

-We have included the significantly enriched eCLIP peaks in the revised Fig. 2h and Supplementary. Fig 7. Since some genes contain dozens of peaks (e.g. >60 peaks on *NEAT1* transcript, Fig. 1c), it may not be feasible for us to include all peaks; however, we have mentioned in the figure legends that the detailed information about all significantly enriched peaks can be found in Supplementary table 1.

3. Lines 101-103: No presence of others RBPs in the DAP3 eCLIP elutes confirmed the specificity of the DAP3 eCLIP-Seq analysis (Supplementary information; Supplementary Fig. 1a and b). This is far too strongly stated. As mentioned in the initial review, the standard method in the field to experimentally confirm this degree of specificity of the RBP under study would be to perform RNA visualization (with radiolabeled/fluorescent/biotinylated RNA) in low versus high RNase to show that the recovered RNA is specific to the DAP3 protein size. The supplemental figure referenced is sufficient to validate that the eCLIP is not pulling down the 4 tested proteins, and the overlap with (and overall number of splicing changes seen in) the RNA-seq strongly suggests that the targets seen are real DAP3 interactions, but the strength of this claim needs to be toned down.

-We note the reviewer’s concern here. We have revised our conclusion, as follows:

“No presence of four selected RBPs including SFPQ, NONO, U2AF35 and U2AF65 in the DAP3 eCLIP elutes confirmed that our eCLIP experiment could specifically pull down DAP3-bound RNAs.”

4. The presence of AGGUAAGU motif (Supplementary Fig. 3a), the 5’ splice site consensus sequence in vertebrates, indicates the binding preference of DAP3 for the exon-intron splice junctions. This is probably also too strong – the result indicates that DAP3 has peaks that overlap 5’ splice sites, but ‘binding preference’ would require some sort of binding assay (and as the authors note in Rebuttal

Fig 7, it seems more likely that it's primarily binding to the exon and simply having reads that extend across the 5'ss region).

-Thank the reviewer for the suggestion. We have changed “binding preference” to “DAP3 could bind to the exon-intron splice junctions”.

5. As noted in the first review – the methods section of the manuscript states for eCLIP “Immunoprecipitated using a DAP3 antibody or control IgG”. I appreciate that the authors confirmed in their response that the same antibody listed in other sections was used, but I think the manuscript text itself should contain these details (antibody catalog information, concentration used, secondary beads used, etc.) in that section of the text itself.

-Apologize for missing these details. We have added the antibody information into the “Methods” section. We followed the standard eCLIP protocol and the detailed procedures could be found in the cited publication.

Minor comments:

1. Fig 2j is not referenced in the text

-We thank the reviewer for pointing this out and we have added it to the main text.

Reviewer #2 (Remarks to the Author):

The authors have addressed my prior comments and questions.

-We thank the reviewer for putting so much effort into the paper review and giving valuable suggestions which have significantly improve our manuscript.

References

Graveley, B.R. (2005). Mutually exclusive splicing of the insect Dscam pre-mRNA directed by competing intronic RNA secondary structures. *Cell* 123, 65-73.

Lovci, M.T., Ghanem, D., Marr, H., Arnold, J., Gee, S., Parra, M., Liang, T.Y., Stark, T.J., Gehman, L.T., and Hoon, S. (2013). Rbfox proteins regulate alternative mRNA splicing through evolutionarily conserved RNA bridges. *Nature structural & molecular biology* 20, 1434-1442.

Pervouchine, D.D., Khrameeva, E.E., Pichugina, M.Y., Nikolaienko, O.V., Gelfand, M.S., Rubtsov, P.M., and Mironov, A.A. (2012). Evidence for widespread association of mammalian splicing and conserved long-range RNA structures. *Rna* 18, 1-15.

Raker, V.A., Mironov, A.A., Gelfand, M.S., and Pervouchine, D.D. (2009). Modulation of alternative splicing by long-range RNA structures in *Drosophila*. *Nucleic acids research* 37, 4533-4544.

Sharma, E., Sterne-Weiler, T., O'Hanlon, D., and Blencowe, B.J. (2016). Global mapping of human RNA-RNA interactions. *Molecular cell* 62, 618-626.

REVIEWERS' COMMENTS

Reviewer #1 (Remarks to the Author):

I appreciate the author's edits and responses, and other than the minor text note below I don't have any remaining comments

>>4. The presence of AGGUAAGU motif (Supplementary Fig. 3a), the 5' splice site consensus sequence in vertebrates, indicates the binding preference of DAP3 for the exon-intron splice junctions. This is probably also too strong – the result indicates that DAP3 has peaks that overlap 5' splice sites, but „binding preference“ would require some sort of binding assay (and as the authors note in Rebuttal

Fig 7, it seems more likely that it's primarily binding to the exon and simply having reads that extend across the 5'ss region).

-Thank the reviewer for the suggestion. We have changed "binding preference" to "DAP3 could bind to the exon-intron splice junctions".

I think there was a miscommunication here – I would suggest something more like "the presence of the AGGUAAGU motif... suggests that DAP3 peaks may include the 5' splice site region", as I think there would need to be additional analysis in this paper to suggest binding to the actual 5' splice site sequence (for that, I think it would be necessary to include at least meta-gene analysis of the exon-intron junction region – do the 'exonic' peaks / does the read density of those peaks in Fig. 1a typically end right at the exon-intron boundary? Or do they extend into the intron? In other words – is this motif actually reflecting the location of DAP3 crosslinking & binding or reflecting a property of peak width extending through the 5' splice site?)

REVIEWERS' COMMENTS

Reviewer #1 (Remarks to the Author):

I appreciate the author's edits and responses, and other than the minor text note below I don't have any remaining comments

>>4. The presence of AGGUAAGU motif (Supplementary Fig. 3a), the 5' splice site consensus sequence in vertebrates, indicates the binding preference of DAP3 for the exon-intron splice junctions. This is probably also too strong – the result indicates that DAP3 has peaks that overlap 5' splice sites, but „binding preference“ would require some sort of binding assay (and as the authors note in Rebuttal Fig 7, it seems more likely that it's primarily binding to the exon and simply having reads that extend across the 5' splice region).

-Thank the reviewer for the suggestion. We have changed “binding preference” to “DAP3 could bind to the exon-intron splice junctions”.

I think there was a miscommunication here – I would suggest something more like “the presence of the AGGUAAGU motif... suggests that DAP3 peaks may include the 5' splice site region”, as I think there would need to be additional analysis in this paper to suggest binding to the actual 5' splice site sequence (for that, I think it would be necessary to include at least meta-gene analysis of the exon-intron junction region – do the ‘exonic’ peaks / does the read density of those peaks in Fig. 1a typically end right at the exon-intron boundary? Or do they extend into the intron? In other words – is this motif actually reflecting the location of DAP3 crosslinking & binding or reflecting a property of peak width extending through the 5' splice site?)

-- We thank the reviewer for the clarification and further suggestion. We have analysed the DAP3 eCLIP peak coverage at -50 to +50 nt region surrounding 5' and 3' splice sites (figure below, also added as supplementary fig3a). A subset of the DAP3 eCLIP peaks was found to extend across the 5' splice site to the intron region, explaining why the 5' splice site consensus sequence was detected as one of the most enriched DAP3 binding motifs. Since our eCLIP experiment utilized the UV crosslinking and removed un-crosslinked RNA fragments by sonication and RNase I digestion, we believe that this motif indeed reflects the DAP3 binding site extending across the 5' splicing site to the intron.

Figure: DAP3 eCLIP peak coverage at -50 to 50 nucleotides (nt) around 5' and 3' splice sites.